# HIDDEN SCHEMA NETWORKS

## ABSTRACT

Starting from the assumption that a large proportion of semantic content is necessarily relational, we introduce a neural language model that discovers networks of symbols (schemata) from text datasets. Using a variational autoencoder (VAE) framework, our model encodes sentences into sequences of symbols (composed representation), which correspond to the nodes visited by biased random walkers on a global latent graph. We first demonstrate that the model is able to uncover ground-truth graphs from artificially generated datasets of random token sequences. Next, we leverage *pretrained* BERT and GPT-2 language models as encoder and decoder, respectively, to train our model on language modelling and commonsense knowledge generation tasks. Qualitatively, the model is able to infer schema networks whose nodes (symbols) can be interpreted as encoding different aspects of natural language (as e.g. topics or sentiments). Quantitatively, our results show that the model successfully interprets the inferred symbol sequences, as it achieves state-of-the-art scores on language modeling benchmarks. Source code to reproduce all experiments is provided with the supplementary material.

## 1 INTRODUCTION

Much of the developmental and causal theories of human cognition are predicated on *relational* structures of knowledge that naturally exhibit *compositionality*. Semantic content is intrinsically relational, as one is only able to explain a given unit of knowledge – such as a concept, word or perception – insofar as there are other units of knowledge which relate to it (Block, 1986). Thus we can partially construe a concept through its relationships to other concepts (like when we say "a dog is an animal that barks"), just as we can partially construe it through its relationships to our perceptions (when we say "*that* is a dog", whilst pointing to a dog on the street) or the words we use (when we use the word *dog* to refer to the concept *dog*). Likewise, we can partially construe words not only through their relationships to concepts or percepts, but also through their relationships to other words, as words that occur in the same context tend to have similar meanings (Harris, 1954; Firth, 1957). Note that is precisely this contextual semantic content of words what we have explicit access to, when processing our raw text datasets. On the other hand, generalization, reasoning and understanding seem to be inevitably tied to the compositional nature of knowledge. Indeed, the ability to compose a set of knowledge units (and their relations) into new, more complex relational units, which can be deployed to understand and reason about unseen data – a feature usually referred to as combinatorial generalization – is regarded as key to human-level intelligence (Fodor & Pylyshyn, 1988; Fodor & Lepore, 2002; Lake et al., 2017; Battaglia et al., 2018). Relational structures allowing for compositionality thus seem to comprise not a sufficient, but a necessary attribute of any representation scheme that strives for the generalization power of human cognition.

From the computational side, if one is to inform any modern machine learning model with such structural characteristics, one will initially encounter the problem of finding suitable primitives or data structures. In natural language processing (NLP), for example, it has become common place to leverage distributed *continuous representations* of words (Bengio et al., 2003) for different downstream tasks. Such representations are trained to encode average contextual semantics – precisely the kind of semantic content typical of word co-occurrence relations we mentioned above – into a semantic space, which allows meaning to change continuously within it (Mikolov et al., 2013). Yet, despite earlier attempts (Mitchell & Lapata, 2008), it is unclear whether such representations can be meaningfully composed into representations of, say, unseen sentences and thus mimic the compositional character of natural language. More recently, contextualized continuous word

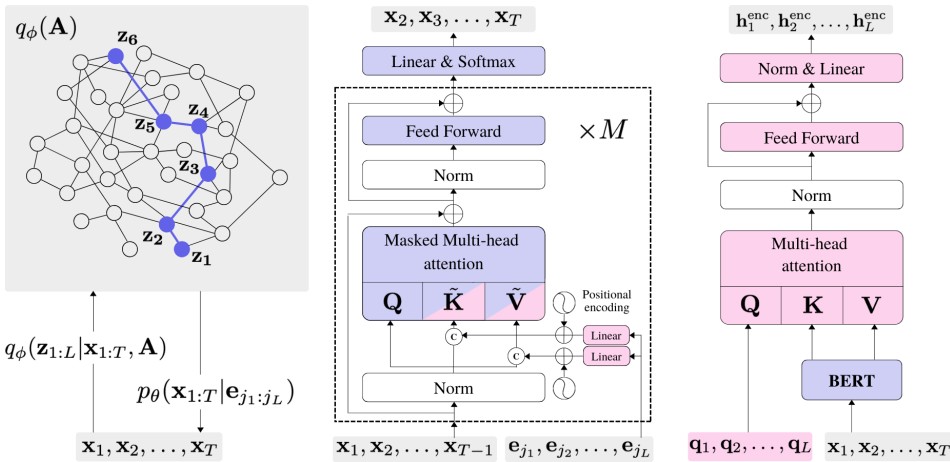

Figure 1: **Left:** Diagram of Hidden Schema Network model. **Center:** Decoder architecture as a modified GPT-2 model of $M$ layers, with a pseudo-self-attention mechanism to attend to the schema $\mathbf{e}_{j_1:j_L}$. Please see the Appendix for details. The "c" operations labels concatenation. **Right**: Encoder architecture as BERT model, followed by a single Transformer block. In both center and right figure purple shaded blocks represent submodules with pretrained parameters. Pink shaded blocks represent submodules with randomly initialized parameters.

representations inferred by deep learning architectures have shown spectacular results in many NLP tasks (Radford et al., 2018; Devlin et al., 2018; Radford et al., 2019; Brown et al., 2020). Their success stems from those models' ability to infer flexible representations through, *inter alia*, raw, massive datasets, data-scalable attention mechanisms and minimal inductive biases (Vaswani et al., 2017). These representations are known to not only contain rich contextual word semantics, but also consistently encode sentence-level grammar (Hewitt & Manning, 2019), and the models from which they are obtained seem to implement some notions of compositionality too (Hupkes et al., 2020; Wei et al., 2022). Nevertheless, it is still unclear whether such representations can be composed into representations of novel sentences (Yu & Ettinger, 2020; Bhathena et al., 2020). In fact, most of their syntactic properties are implicit and therefore inferred only a posteriori, typically through probes which neither guarantee their presence, nor establish how they were obtained in the first place (Rogers et al., 2020).

In this work we use a VAE framework (Kingma & Welling, 2013; Rezende et al., 2014) to develop a language model – the Hidden Schema Network model (HSN) – that enforces, via inductive biases, a discrete, *relational* structure for sentence representation which allows for *compositionality*[1], while exploiting the well-known advantages of attention models and contextualized, pretrained representations. We first demonstrate that the model is able to uncover ground-truth graphs from artificially generated datasets of random token sequences. Next, we leverage our methodology to translate the implicit lexical and grammatical aspects of language encoded by *pretrained* BERT and GPT-2 language models into explicit relational structures, and apply the latter on language modelling and commonsense knowledge generation tasks.

Our main contribution is then an exploration of a novel way to integrate discrete (symbols), relational (graphs) and continuous (neural representations) machine learning components into an end-to-end, differentiable representation learning algorithm for natural language modelling. Our aim is thus to try to connect the modern NLP paradigm with classical notions of linguistics, and begin to answer the recent calls for neuro-symbolic integration (Garcez & Lamb, 2020; Cartuyvels et al., 2021).

---

[1]Note that throughout the paper we refer only to *compositionality of representations* and not to the compositional functions that can be implemented by the models we use. The latter, functional compositionality, is studied by e.g. Hupkes et al. (2020).

## 2 RELATED WORK

In cognitive psychology, a schema is (roughly) defined as a large, complex unit of knowledge representing what is typical of a group of instances (Bartlett, 1932; Piaget, 1948; Rumelhart, 2017). Marvin Minsky's frames (Minsky, 1974; 1975) are similar in function to a schema, but perhaps more easily characterized in terms of data structures. We use these terms in a loose fashion, however. Our aim being only to be suggestive of the general problem of knowledge representation (Thagard, 1984). We are in fact concerned with representation schemes for natural language processing. Within the context of linguistics, Jackendoff (1978) argues that there must be a level of representation – the so-called conceptual structures – at which information conveyed by language must be compatible with information coming from sensory systems. Conceptual structures must, he goes on, be able to represent all the conceptual distinctions made by natural language, and provide some degree of compositionality. Earlier computational models implementing (some kind of) conceptual structure rely on either hand-coded (semantic) network representations (Quillan, 1966; Collins & Quillian, 1969; Brachman, 1977) or hand-coded databases (McClelland & Rogers, 2003). Other works focus instead on learning semantic representations directly from text data via topic models (Griffiths et al., 2007b), and even infer latent concept graphs through nonparametric priors (Chambers et al., 2010).

In sharp contrast with these works, modern, neural-based language models incorporate no explicit linguistic notions, and leverage massive datasets and attention mechanisms, in the form of large pre-trained language models and contextualized, continuous word representations. We build on top of these ideas, while trying to connect back with models of conceptual structure, which necessarily involve discrete representations (van den Oord et al., 2017; Hu et al., 2017; Zhao et al., 2018; Kaiser & Bengio, 2018; Kaiser et al., 2018).

## 3 HIDDEN SCHEMA NETWORKS

We address the problem of learning the joint probability distribution over sequences of words, while inferring interpretable representations capturing their semantics. Neural autoregressive language models approximate such distributions with a product over conditional probabilities, such that

$$p(\mathbf{x}_{1:T}) = \prod_{i=1}^{T} p_\theta(\mathbf{x}_i | \mathbf{x}_{<i}), \tag{1}$$

where $\mathbf{x}_{1:T} = (\mathbf{x}_1, \mathbf{x}_2, \ldots, \mathbf{x}_T)$ labels the sequence of words in question, and each conditional is given by (the pdf of) a categorical distribution over some vocabulary of size $V$. The class probabilities of these conditionals are generally computed as $\boldsymbol{\pi}_i = \text{softmax}(\mathbf{W} \cdot \mathbf{h}_\theta(\mathbf{x}_{<i}))$, with $\mathbf{W} \in \mathbb{R}^{V \times D}$ trainable, and $D$ the output dimension of $\mathbf{h}_\theta$, a deep neural network model with parameter set $\theta$ (Bengio et al., 2003). Models of this form allow for tractable estimation of and sampling from either the joint distribution, or any product of the conditionals in Eq. 1. Indeed, their recent implementation in terms of large-capacity, self-attention architectures such as GPT-2 (Radford et al., 2019) has been shown to generate syntactically correct, diverse and fluent text. Yet, most of the linguistic structure encoded by the output representations of these models is implicit and difficult to interpret (Rogers et al., 2020). In what follows we shall condition the joint distribution of Eq. 1 on an additional latent, *discrete* representation which can, at least in principle, capture the relational and compositional features of semantic content.

Let us assume there is a set $\mathcal{E} = \{\mathbf{e}_1, \mathbf{e}_2, \ldots, \mathbf{e}_K\}$ of $K$ symbols that encode some high-level, abstract semantic content of natural language. Let this set be the set of nodes of a hidden (semantic) graph $\mathcal{G}$, with adjacency matrix $\mathbf{A}$, so that adjacent (connected) symbols are semantically related. These symbols can generically be defined as learnable, dense vectors in $\mathbb{R}^S$, for some dimension $S$. Without loss of generality, however, we opt below for simple indicator ("one-hot") vectors of dimension $K$ instead. We define a *schema* $\mathbf{e}_{j_1:j_L}$ as a sequence of $L \ll K$ symbols $(\mathbf{e}_{j_1}, \mathbf{e}_{j_2}, \ldots, \mathbf{e}_{j_L})$, where the indices $j_1, \ldots, j_L$ label a subset of connected nodes in $\mathcal{G}$. Accordingly, we refer to $\mathcal{G}$ as a *schema network*. The symbols composing the schemata are chosen through a $L$-step stochastic process conditioned on $\mathcal{G}$. Partially motivated by research on random walks and human memory search (Griffiths et al., 2007a; Abbott et al., 2012), as well as by the simplicity of their inference, we choose to compose the schemata via biased random walk processes on $\mathcal{G}$, and leave exploring different schema processes for future work . Let us now specify the generative model in detail.

## 3.1 GENERATIVE MODEL

We write the joint probability over a sequence $\mathbf{x}_{1:T}$ of $T$ words, together with the hidden graph $\mathcal{G}$, as

$$p_\theta(\mathbf{x}_{1:T}, \mathbf{A}) = \sum_{\mathbf{z}_{1:L}} p_\theta(\mathbf{x}_{1:T}|\mathbf{e}_{j_1:j_L})p(\mathbf{z}_{1:L}|\mathbf{A})p(\mathbf{A}), \qquad (2)$$

where $\mathbf{z}_{1:L}$ labels the sequence $\mathbf{z}_1, \ldots, \mathbf{z}_L$ of $K$-dimensional, one-hot vectors representing the node labels $j_1, \ldots, j_L$ visited by a random walker on $\mathcal{G}$, and $\theta$ denotes the trainable model parameters. Note that we introduced the one-hot representation of $j_i$ for notational convenience, as shall become evident below[2]. Next, we specify the different components of Eq. 2.

**Prior over (global) graph**. A prior on the adjacency matrix $p(\mathbf{A})$ allows us to control the topological properties of $\mathcal{G}$. One can choose, for example, random graph models whose degree distribution asymptotically follow a power law (Barabási & Albert, 1999), or unbiased, maximum entropy graph models, with respect to some given constrains (Park & Newman, 2004). For the sake of simplicity we choose a Bernoulli (Erdös-Rényi) random graph model (Solomonoff & Rapoport, 1951; Erdös & Rényi, 1959), for which each link $a_{ij}$ is defined via an independent Bernoulli variable with some fixed, global probability $p \in [0, 1]$, so that

$$p(\mathbf{A}) = \prod_{i,j=1}^{K} p^{a_{ij}} (1-p)^{1-a_{ij}} . \qquad (3)$$

The probability $p$ will be a hyperparameter of our model.

**Prior over random walks**. The probability $p(\mathbf{z}_{1:L}|\mathbf{A})$ of a random walk over the nodes of $\mathcal{G}$ can generally be written as

$$p(\mathbf{z}_{1:L}|\mathbf{A}) = p(\mathbf{z}_1)\prod_{i=2}^{L} p(\mathbf{z}_i|\mathbf{z}_{i-1}, \mathbf{A}) = \left(\prod_{m=1}^{K} \rho_m^{z_1^m}\right)\prod_{i=2}^{L}\left(\prod_{j=1}^{K}\prod_{k=1}^{K} P_{k,j}^{z_i^k z_{i-1}^j}\right), \qquad (4)$$

where $p(\mathbf{z}_1)$ labels the probability of selecting $j_1$ as the starting point of the walk, and it is given by (the pdf of) a categorical distribution over the nodes of $\mathcal{G}$, with class probabilities $\{\rho_i\}_{i=1}^{K}$. Similarly $p(\mathbf{z}_i|\mathbf{z}_{i-1}, \mathbf{A})$ labels the conditional probability of jumping from $j_{i-1}$ to $j_i$, which we define in terms of a $K \times K$ transition probability matrix $\mathbf{P}$. Now, to allow for biased random walks, let each node $k$ on $\mathcal{G}$ be given a positive weight $f_k$, so that the probability of jumping from $j$ to $k$ is proportional to $f_k A_{kj}$. We then write the transition probability matrix as

$$P_{k,j} = \frac{f_k A_{kj}}{\sum_{i=1}^{K} f_i A_{ij}}, \qquad (5)$$

so that the motion of the random walker is biased according to the node weights $f_k$. These weights should be understood as encoding aspects of the diffusion dynamics that are independent of the topology of the graph (Gómez-Gardenes & Latora, 2008; Lambiotte et al., 2011). Three comments are in order: first, note that one can also *train* the prior over walks by making the vectors $\boldsymbol{\rho}$ and $\mathbf{f}$ learnable. Second, setting the node weights $\mathbf{f} = \mathbb{I}$ and the class probabilities $\boldsymbol{\rho} = \frac{1}{K}\mathbb{I}$, with $\mathbb{I}$ the $K$-dimensional vector of ones, yields a *uniform* random walk over $\mathcal{G}$, i.e. a process in which the walker has equal probability of jumping to any of its neighbors. Third, one can also allow for *inhomogeneous* random walks in which the probability matrix changes at each step of the random walk. Such processes can be parameterized with a sequence of weights $\mathbf{f}^{[1]}, \mathbf{f}^{[2]}, \ldots, \mathbf{f}^{[L-1]}$.

**Decoder and likelihood**. Just as in Eq. 1, we define the joint probability over word sequences as a product of conditional probabilities, this time conditioned on the schema $\mathbf{e}_{j_1:j_L}$ too, that is

$$p_\theta(\mathbf{x}_{1:T}|\mathbf{e}_{j_1:j_L}) = \prod_{i=1}^{T} p_\theta(\mathbf{x}_i|\mathbf{x}_{<i}, \mathbf{e}_{j_1:j_L}), \quad \boldsymbol{\pi}_i = \text{softmax}(\mathbf{W} \cdot \mathbf{h}_\theta^{\text{dec}}(\mathbf{x}_{<i}, \mathbf{e}_{j_1:j_L})), \qquad (6)$$

with $\boldsymbol{\pi}_i$ the class probabilities of the $i$th conditional, $\mathbf{W} \in \mathbb{R}^{V \times D}$ trainable, and $\mathbf{h}_\theta^{\text{dec}}$ a deep neural network model. We let $\mathbf{h}_\theta^{\text{dec}}$ be a *pretrained* GPT-2 language model, and modify it to also process the

---

[2]Explicitly, $j_i$ denotes the index of the non-zero component of $\mathbf{z}_i$, i.e. $j_i = \{k \in [1, K] : z_i^k = 1\}$, with the superindex $k$ denoting the components of $\mathbf{z}_i$.

schema $\mathbf{e}_{j_1:j_L}$, but remark that any other model for sequence processing (as e.g. a recurrent neural net) could be used instead. A bit more in detail, to condition GPT-2 on $\mathbf{e}_{j_1:j_L}$, without perturbing its optimized weights too much, we use the *pseudo-self*-attention (PSA) mechanism introduced by Ziegler et al. (2019). In a nutshell, this mechanism augments the key and value matrices of GPT-2 in their first $L$ rows with projections of $\mathbf{e}_{j_1:j_L}$. Figure 1 shows an illustration of the complete decoder model, including the PSA mechanism. Please check Appendix A for the explicit equations of the latter.

## 3.2 INFERENCE MODEL

The generative model we presented above is hierarchical. The random graph is shared across all sentences and thus constitutes a global latent object. The random walks, in contrast, are local random variables. Our task is to infer the schema and graph posterior distributions that best describe the collection of word sequences in our dataset. To do this, we approximate the true posterior distribution of these variables with a variational posterior of the form

$$q_\phi(\mathbf{z}_{1:L}, \mathbf{A}|\mathbf{x}_{1:T}) = q_\phi(\mathbf{z}_{1:L}|\mathbf{x}_{1:T}, \mathbf{A})q_\phi(\mathbf{A}), \tag{7}$$

where $\phi$ labels the set of trainable parameters. Let us specify each of its components.

**Posterior over (global) graph**. We model the posterior over the graph assigning again Bernoulli variables to its links, but we let the probability of observing each link depend on the global symbols

$$q_\phi(\mathbf{A}) = \prod_{i,j} p_\phi(\mathbf{e}_i, \mathbf{e}_j)^{a_{ij}} \left(1 - p_\phi(\mathbf{e}_i, \mathbf{e}_j)\right)^{1-a_{ij}}, \text{ where } p_\phi(\mathbf{e}_i, \mathbf{e}_j) = \text{sigmoid}(g_\phi(\mathbf{e}_i, \mathbf{e}_j)), \tag{8}$$

with $g_\phi : \mathcal{E} \times \mathcal{E} \to \mathbb{R}$ a deep neural network, and $p_\phi(\mathbf{e}_i, \mathbf{e}_j) \in [0, 1]$, for all $\mathbf{e}_i \in \mathcal{E}$, the link probabilities. Our reasoning here is that the network $g_\phi$ should infer graphs connecting symbols which are semantically related via the encoded sentences.

**Posterior over random walks (encoder model)**. Analog to Eq. 4 we model the posterior probability over random walks on $\mathcal{G}$ as

$$q_\phi(\mathbf{z}_{1:L}|\mathbf{x}_{1:T}, \mathbf{A}) = \left(\prod_{i=1}^{K} \rho_i(\mathbf{x}_{1:T}, \phi)^{z_1^i}\right) \prod_{i=2}^{L} \left(\prod_{j=1}^{K} \prod_{k=1}^{K} \left(Q_{k,j}^{[i-1]}(\mathbf{x}_{1:T}, \mathbf{A}, \phi)\right)^{z_i^k z_{i-1}^j}\right), \tag{9}$$

where instead of having a single transition probability matrix, we have a sequence of them, thereby allowing the posterior to capture inhomogeneous random walks. Note that we could have also chosen a mean-field decomposition along the steps of the random walk, simply by either ignoring the dependency on the graph, or making the graph fully connected (see Appendix B.4). Going back to Eq. 9, we model the probabilities over the starting point of the random walks and the transition matrices as follows

$$\boldsymbol{\rho}(\mathbf{x}_{1:T}, \phi) = \text{softmax}(\mathbf{h}_1^{\text{enc}}), \tag{10}$$

$$Q_{k,j}^{[i]}(\mathbf{x}_{1:T}, \mathbf{A}, \phi) = \frac{f_k^{[i]}(\mathbf{x}_{1:T}, \phi) A_{kj}}{\sum_m f_m^{[i]}(\mathbf{x}_{1:T}, \phi) A_{mj}}, \text{ with } \mathbf{f}^{[1]}, \dots, \mathbf{f}^{[L-1]} = \exp(\mathbf{h}_{2:L}^{\text{enc}}), \tag{11}$$

where $\mathbf{h}_1^{\text{enc}}, \mathbf{h}_2^{\text{enc}}, \dots, \mathbf{h}_L^{\text{enc}} \in \mathbb{R}^D$ is the sequence of outputs of a deep neural network model $\mathbf{h}_\phi^{\text{enc}}(\mathbf{x}_{1:T})$ processing the input sequence of $T$ words. The model $\mathbf{h}_\phi^{\text{enc}}(\mathbf{x}_{1:T})$ must then map a sequence of $T$ vectors to a sequence of $L$ vectors. We define $\mathbf{h}_\phi^{\text{enc}}$ by a *pretrained* BERT model (Devlin et al., 2018), followed by a single Transformer block, randomly initialized. The Transformer block processes the $T$ ($D$-dimensional) outputs from BERT as keys and values, together with a set of $L$ learnable vectors $\mathbf{q}_{1:L}$ as queries. The right hand side of Figure 1 illustrates the complete encoder architecture.

## 3.3 TRAINING OBJECTIVE

To optimize the parameter sets $\{\theta, \phi\}$ of our latent variable model we would, as usual, maximize a variational lower bound on the logarithm of the marginal likelihood $p_\theta(\mathbf{x}_{1:T})$ (Bishop, 2006). It is, however, well known that VAE models tend to encounter problems learning representations encoding information about the data – the so-called posterior collapse problem – especially when dealing with

| Graph $\mathcal{G}^*$ | ROC AUC | $||\mathcal{G}^* - \mathcal{G}||_F$ | $||\mathcal{G}^{\text{rand}} - \mathcal{G}||_F$ | N. edges($\mathcal{G}$) | N. edges($\mathcal{G}^*$) |
|---|---|---|---|---|---|
| Barabasi | $0.989 \pm 0.001$ | $17 \pm 2$ | $26 \pm 1$ | $1360 \pm 104$ | 291 |
| Erdos | $0.94 \pm 0.06$ | $36.8 \pm 0.8$ | $44 \pm 2$ | $3131 \pm 156$ | 2092 |

Table 1: Inference of ground-truth random graphs

natural language (Bowman et al., 2015). To solve this issue practitioners resort to maximizing the variational lower bound, together with the mutual information between data and representations (Zhao et al., 2018; Fang et al., 2019; Zhao et al., 2019). We follow this same route and show (in Appendix B.3) that maximizing the lower bound and the mutual information corresponds to maximizing the objective

$$
\mathcal{L}[\theta, \phi] = \frac{1}{N} \sum_{n=1}^{N} \mathbb{E}_{q_\phi(\mathbf{z}_{1:L}|\mathbf{x}_{1:T}^{(n)}, \mathbf{A}) q_\phi(\mathbf{A})} \log p_\theta(\mathbf{x}_{1:T}^{(n)}|\mathbf{z}_{1:L})
$$
$$
- \mathbb{E}_{q_\phi(\mathbf{A})} \text{KL}\Big[ q_\phi^*(\mathbf{z}_{1:L}|\mathbf{A}); p(\mathbf{z}_{1:L}|\mathbf{A}) \Big] - \text{KL}[q_\phi(\mathbf{A}); p(\mathbf{A})], \quad (12)
$$

where KL labels the Kullback-Leibler divergence (Kullback & Leibler, 1951) between prior and posterior distributions, and $q_\phi^*(\mathbf{z}_{1:L}|\mathbf{A})$ is the aggregated posterior distribution over random walks. The latter is defined as $\mathbb{E}_{p_\mathcal{D}(\mathbf{x}_{1:T})}[q_\phi(\mathbf{z}_{1:L}|\mathbf{x}_{1:T}, \mathbf{A})]$ and is in general intractable. In practice, we approximate it with an expression identical to Eq. 9, but with the class probabilities and transition matrices (Eqs. 10 and 11) replaced with their data-averaged counterparts. We refer the reader to Appendix B for details on this, as well as for the explicit, closed-form expressions of the Kullback-Leibler terms in Eq. 12. The full training algorithm is presented in Appendix C.

## 4 PROOF OF CONCEPT: INFERRING GROUND-TRUTH RANDOM GRAPHS

Before testing the behaviour of our methodology on natural language data, we evaluate the ability of the model to infer hidden graph structures from sequential data in a controlled experiment. To this end, we define a synthetic language model with an underlying, ground-truth graph $\mathcal{G}^*$ as follows: Given a graph $\mathcal{G}^*$ with $K$ nodes, and a vocabulary of random tokens $\mathcal{V}$ of size $V$, we assign one random bag of tokens (i.e. one pdf over $\mathcal{V}$) to each node of the graph. Let the $K$ random bags be the $K$ symbols $\{\mathbf{e}_1, \mathbf{e}_2, \dots, \mathbf{e}_K\}$ of the synthetic language model. We then sample $N$ uniform random walks of length $L$ over $\mathcal{G}^*$, and sample one random token from each symbol (i.e. from each random bag) along the walks. The result is a set of random token sequences of the same length as that of the random walks. Appendix D contains a more detailed description of this generation procedure.

Given this set of random token sequences, the task is to infer the hidden ground-truth graph $\mathcal{G}^*$.

**Experimental settings**. Following the procedure above we generated two datasets from two random graphs with different topologies. One sampled from the Barabási-Albert model (Barabási & Albert, 1999), the other from the Erdös-Rényi model (Erdös & Rényi, 1959). We set both graphs to have $K = 100$ symbols, and the token sequences to have length $L = 10$. Each dataset has a total of $N = 100000$ token sequences. Further details about the random graph model parameters and the dataset statistics can be found in Appendix D. The synthetic datasets are available in the source code.

**A simple proof-of-concept**. We consider a problem in which the set of symbols (random bags) $\mathcal{E}$ is known, so that the ground-truth graph $\mathcal{G}^*$ has a fixed labelling. This setting will allow for simple comparison between $\mathcal{G}^*$ and our inferred graphs. To infer $\mathcal{G}^*$ we used a simplified version of HSN, namely: we (i) replace BERT in Fig. 1 with a 2-block Transformer encoder (Vaswani et al., 2017); (ii) set the graph model $g_\phi$ (Eq. 8) to a single-layer, feed forward network; and (iii) note that, since the symbols are known, the likelihood of the model is simply given by $\prod_{i=1}^{L} \mathbf{e}_{j_i}$ where, as before, $j_i$ denotes the index of the non-zero component of $\mathbf{z}_i$. We train this model by maximizing Eq. 12 and refer to Appendix D for details on hyperparameters, training procedure and model sizes.

**Results**. Table 1 shows our results for our two synthetic datasets. Specifically, we compute the Area Under the Receiver Operating Characteristic Curve (ROC AUC) of our model $q_\phi(\mathbf{A})$ with respect to $\mathcal{G}^*$, and the Frobenious norm between $q_\phi$ and two graphs: the ground-truth one $\mathcal{G}^*$, and a second random graph $\mathcal{G}^{\text{rand}}$ sampled from the same random graph model as $\mathcal{G}^*$. We train ten (10) models

| Method | PTB | YAHOO | YELP |
|---|---|---|---|
| GPT2 [one epoch] | 24.23 | 22.00 | 23.40 |
| GPT2 [fine-tuned] | 19.14 | 20.64 | 19.77 |
| iVAE$_{MI}$ | 53.44 | 47.93 | 36.88 |
| Optimus | 23.58 | 22.34 | 21.99 |
| HSN $_{(100, 20)}$ | 17.72 | 20.28 | 19.18 |
| HSN $_{(100, 5)}$ | 17.79 | 20.10 | 19.05 |
| HSN $_{(50, 20)}$ | **16.88** | **19.59** | 19.01 |
| HSN $_{(50, 5)}$ | 17.41 | 20.06 | **18.95** |

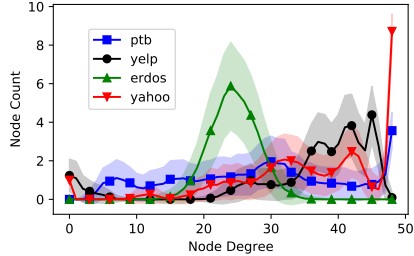

Table 2: Left: Perplexity per word (lower is better) on three datasets. GPT2 [one epoch] and Optimus results were extracted from Li et al. (2020). iVAE$_{MI}$ was taken from Fang et al. (2019). GPT2 [fine-tuned] was computed by us. End-of-sequence tokens are kept during evaluation.

Figure 2: Empirical degree distributions of inferred graphs from each corpora. Results correspond to HSN with $L = 5, K = 50$. We also show the distribution for random graphs with $p = 0.5$. The graphs are sampled 500 times.

in total and display the mean and standard deviation of our results. We also use a different $\mathcal{G}^{\text{rand}}$ for each calculation run. The first metric shows that $q_\phi$ correctly predicts the edges of $\mathcal{G}^*$, whereas the other two metrics show that $\mathcal{G} \sim q_\phi(\mathbf{A})$ is closer to $\mathcal{G}^*$ than to any other random graph sampled from the same distribution. The last two columns in Table 1 show however that $q_\phi(\mathbf{A})$ tends to generate denser graphs as compared to the target.

Having demonstrated that HSN can indeed infer hidden graph structure from sequential data in a simple setting[3], we now move to our main problem: language modelling.

## 5 LANGUAGE MODELLING AND REPRESENTATION LEARNING

Natural language modelling deals with the prediction of the next word in a sentence or document, given a sequence of previously observed words. A natural evaluation metric is therefore the perplexity per word of the model, which is defined as the exponential of the data-averaged, negative log-likelihood of the model, divided by the number of words in the sequence. One complication with this is that latent variable models can only approximately estimate the likelihood function. One can readily see, however, that Eq. 12 is also a lower bound on $\log p_\theta$ (Bishop, 2006) and so, we estimate the perplexity of our models with $\exp(-\mathcal{L}/T)$.

**Datasets and baselines**. We consider three widely used public datasets, namely the Penn Treebank (PTB) (Marcus et al., 1993), Yahoo and Yelp (Yang et al., 2017) corpora. For completeness we include statistics of these datasets in Appendix E. We compare HSN against a pretrained GPT-2, fine-tuned both during a single epoch and until its objective function plateaus. We also compare againts two VAE language models: iVAE$_{MI}$ (Fang et al., 2019) and Optimus (Li et al., 2020). The former implements both encoder and decoder as one-layer LSTMs (Hochreiter & Schmidhuber, 1997). The latter uses pretrained BERT and GPT-2 as encoder and decoder, respectively.

**Experimental settings**. In all experiments we leverage pretrained BERT and GPT-2 models, both with 12 layers, 768 hidden dimensions ($D$) and 12 attention heads. Note that Optimus shares these settings. We use the public HuggingFace implementation of both these models (Wolf et al., 2020). The graph model is set to a 2-layer feed forward network, each with hidden dimension 512, and we also train an inhomogeneous random walk prior model (Eq. 4) by making $\boldsymbol{\rho}$ and the sequence of weights $\mathbf{f}^{[1]}, \mathbf{f}^{[2]}, \ldots, \mathbf{f}^{[L-1]}$ trainable. Furthermore, we explore HSNs with $K = \{50, 100\}$ symbols and hidden random walks of $L = \{5, 20\}$ steps. Let us label these configurations as HSN($K, L$). Additional details on hyperparameters and training procedures can be found in Appendix E.

**Results**. Table 2 shows the perplexity of our model, together with the baselines, evaluated on the test set of the three corpora. HSN achieves a much better performance than all baselines under this metric, which implies it successfully interprets the symbol sequences it uses to encode the sentences. Note in particular that HSNs with 50 symbols perform consistently better than their 100-symbol counterparts.

---

[3]We could, of course, now study the harder problem for which the symbols are unknown. However, the learned graphs model would not be aligned with $\mathcal{G}^*$ making the graph comparison non-trivial.

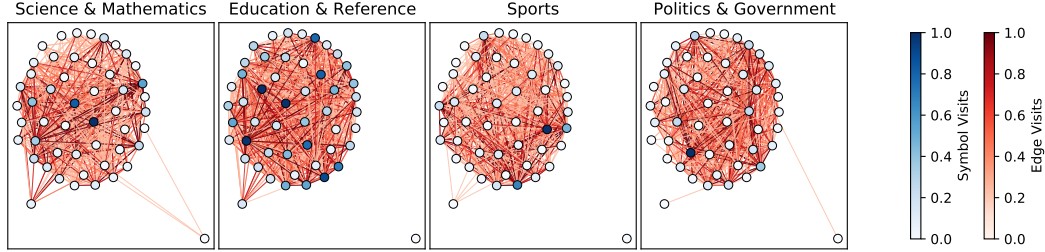

Figure 3: Schema distributions inferred by HSN(50, 5) for four labels of the Yahoo corpora. The node positions in the figure are consistent among labels and were computed using a force-directed embedding of the global graph $\mathcal{G}$.

As we discuss in Appendix E, 100-symbol HSNs tend to infer networks with many disconnected subgraphs, the largest of which has usually about 50 symbols. It appears then that (about) 50 symbols are enough to encode these corpora. We have repeated these experiments five (5) times, each with a different random initialization of the pink shaded blocks in Fig. 1. We find our mean perplexities to be better than all baselines even within error bars. The reader can find these results in Appendix E. To get a deeper insight into the features of these representations we now explore the structure of the learned global graphs $\mathcal{G}$, as well as the semantic content of the schemata.

**Structure of hidden schema networks**. We characterize the structure of $\mathcal{G}$ in terms of five statistics: its (i) diameter $\mathcal{D}$, (ii) average distance $l$, (iii) clustering coefficient $\mathcal{C}$, (iv) number of connected components $\mathcal{CC}$ and (v) degree distribution $P(k)$ (see Appendix E for the definition of these). We report our results in Table 3 for HSN(50, 5). Results for the other HSN configurations can be found in the Appendix, from which we mainly find that longer random walks and larger symbol number generically favor larger $\mathcal{CC}$. Going back to Table 3, we observe that the schema networks from each corpora tend to have smaller average distances $l$, and much larger clustering coefficients $\mathcal{C}$, than any random graphs (with $p = 0.5$) of the same size – where random graphs with $p = 0.5$ correspond to our prior model. Let us remark that the combinations of these two features defines the so-called *small-world structure* (Watts & Strogatz, 1998). Intuitively, a larger $\mathcal{C}$ implies that a random walker starting from a given node $k$ will have a larger number of paths bringing it back to $k$. In such an scenario, random walkers tend to cluster in neighborhoods around their starting point – a property that could help encode different semantic aspects in different regions of $\mathcal{G}$. Another consequence is that one could expect schemata composed of repeated symbols. Figure 2 shows the degree distributions of HSN(50, 5). Here we see another aspect on which the schema networks differ from a purely random graph. In particular, the former are more densely connected than the latter.

**Schemata and semantics**. To qualitatively grasp the semantic content of the learned schemata we take advantage of the labels available to both Yahoo and Yelp corpora. For example, Figure 3 displays the random walk distributions over the schema networks for four (4) subsets of Yahoo, as inferred with HSN(50, 5). Similar plots for all subsets (labels) of both corpora, extracted with all our HSN configurations can be found in Appendix E. Note how the "hot" symbols per category reside on different regions of the graphs – as suspected already from the large clustering coefficient of $\mathcal{G}$ – and yet, the "Science & Math" schemata (both nodes and edges) of Yahoo *closer* to the "Education & Reference" schemata than to the "Sport" schemata, where closer nodes in the figure indicate well-connected nodes in the underlying graph $\mathcal{G}$. We can understand these findings as indicating that the schemata indeed encode semantic notions of their corpora. A similar picture

| Dataset | n. edges | $\mathcal{D}$ | $l$ | $\mathcal{C}$ | $\mathcal{CC}$ | largest $\mathcal{CC}$ |
|---|---|---|---|---|---|---|
| PTB | $694.26 \pm 9.47$ | $2.00 \pm 0.00$ | $1.43 \pm 0.01$ | $0.83 \pm 0.01$ | $1.00 \pm 0.00$ | $50.00 \pm 0.00$ |
| YAHOO | $892.67 \pm 8.22$ | $2.00 \pm 0.00$ | $1.24 \pm 0.01$ | $0.84 \pm 0.00$ | $2.00 \pm 0.04$ | $49.00 \pm 0.04$ |
| YELP | $891.06 \pm 6.50$ | $2.73 \pm 0.46$ | $1.24 \pm 0.03$ | $0.84 \pm 0.01$ | $2.24 \pm 0.85$ | $48.76 \pm 0.85$ |
| Random | $611.69 \pm 17.61$ | $2.00 \pm 0.00$ | $1.50 \pm 0.01$ | $0.50 \pm 0.02$ | $1.00 \pm 0.00$ | $50.00 \pm 0.00$ |

Table 3: Statistics of Schema Networks per corpora with $K = 50$ and $L = 5$. Random denotes an Erdös-Rényi model with $p = 0.5$ for the corresponding $K$.

| | COMET(GPT2) | COMET(GPT2-XL) | COMET (BART) | HSN | HSN[prior] | HSN[KD] |
|---|---|---|---|---|---|---|
| BLEU-2 | 0.225 | 0.300 | 0.330 | **0.332** | 0.067 | 0.125 |
| BERT Score | 0.486 | 0.638 | 0.650 | **0.782** | 0.435 | 0.561 |

Table 4: Metrics of object generation quality for ATOMIC dataset. COMET(GPT2-XL) and COMET (BART) results were extracted from Hwang et al. (2020). COMET(GPT2-XL) was computed by us. The HSN models have $K = 50, L = 20$. All models use greedy decoding for *all* text prefixes in the dataset.

holds for Yelp. Finally, we have also defined and explored "schema interpolations" (Appendix E) and have investigated how the schemata are attended to by the model (Appendix F). These experiments (qualitatively) show too that the schemata encode different semantic notions of natural language.

## 6 COMMONSENSE REASONING GENERATION

It has been proposed recently that large, pretrained language models fine-tuned on (natural language) knowledge graph (KG) tuples, can express their encoded knowledge through language generation, thereby providing commonsense knowledge on demand (Bosselut et al., 2019; Hwang et al., 2020). These commonsense KGs live however in data (i.e. text) space – the nodes and edges are represented by either single words or sequences of them. This observation led us to investigate whether one could use the COMET framework of Bosselut et al. (2019), together with the inductive biases of HSN, to translate the implicit knowledge of pre-trained models into KGs *in representation space*. Arguably so abstract a KG could encompass larger commonsense KGs in data space. With this intuition in mind, let us revisit the COMET framework.

**Task, datasets and baselines**. Consider a training KG of natural language tuples of the form $(\mathbf{s}, \mathbf{r}, \mathbf{o})$, where $\mathbf{s} = (\mathbf{x}_1^s, \ldots, \mathbf{x}_{|\mathbf{s}|}^s)$ labels the *phrase subject* of the tuple, $\mathbf{r} = \mathbf{x}^r$ is the *relation token* and $\mathbf{o} = (\mathbf{x}_1^o, \ldots, \mathbf{x}_{|\mathbf{o}|}^o)$ is the *phrase object* of the tuple. The task is to generate the object $\mathbf{o}$, given $\mathbf{s}$ and $\mathbf{r}$. In other words, to infer the distribution $p(\mathbf{o}|[\mathbf{s}, \mathbf{r}])$. COMET finetunes its pretrained models by maximizing the likelihood of the object, conditioned on the sequence $[\mathbf{s}, \mathbf{r}] = (\mathbf{x}_1^s, \ldots, \mathbf{x}_{|\mathbf{s}|}^s, \mathbf{x}^r)$ (Bosselut et al., 2019). In contrast, HSN is trained to auto-encode the complete sequence $[\mathbf{s}, \mathbf{r}, \mathbf{o}] = (\mathbf{x}_1^s, \ldots, \mathbf{x}_{|\mathbf{s}|}^s, \mathbf{x}^r, \mathbf{x}_1^o, \ldots, \mathbf{x}_{|\mathbf{o}|}^o)$ and is evaluated on object generation tasks, conditioned not only on $[\mathbf{s}, \mathbf{r}]$ but also on the schema $\mathbf{e}_{j_1:j_L}$. For this preliminary study we focus on the ATOMIC dataset (Sap et al., 2019a), evaluate the quality of the generated objects with both, BLEU-2 (Papineni et al., 2002) and BERT Score (Zhang* et al., 2020) metrics, and compare against GPT-2, GPT-2-XL and BART, all trained within the COMET framework(Hwang et al., 2020).

**Results.** Table 4 shows HSN outperforms all baselines[4], which entails it successfully infers and interprets schemata encoding the KG tuples. These schemata, however, are inferred via a posterior of the form $q_\phi(\mathbf{z}_{1:\frac{L}{2}}|[\mathbf{s}, \mathbf{r}], \mathbf{A})q_\phi(\mathbf{z}_{\frac{L}{2}+1:L}|[\mathbf{s}, \mathbf{r}, \mathbf{o}], \mathbf{z}_{\frac{L}{2}}, \mathbf{A})$ – see Appendix G for details. Yet, in practice, one does not have access to any object during *inference*. The classical solution, à la Kalman Filter, is to replace $q_\phi(\mathbf{z}_{\frac{L}{2}+1:L}|[\mathbf{s}, \mathbf{r}, \mathbf{o}], \mathbf{z}_{\frac{L}{2}}, \mathbf{A})$ with a local prior model of the form $p_\theta(\mathbf{z}_{\frac{L}{2}+1:L}|[\mathbf{s}, \mathbf{r}], \mathbf{z}_{\frac{L}{2}}, \mathbf{A})$, and train the latter via the KL term in Eq. 12. Maximizing the mutual information, however, averages out all local information from the prior and hinders its learning – see e.g. HSN[prior] in Table 4. An alternative is to train, in the spirit of knowledge distillation (Hinton et al., 2015), a third-party model on the inferred schemata, to predict $\mathbf{z}_{\frac{L}{2}+1:L}$ conditioned on $\mathbf{z}_{1:\frac{L}{2}}$. Our preliminary results, reported as HSN[KD] in Table 4, improve upon HSN[prior] and even outperform COMET(GPT2) in the BERT Score.

## 7 CONCLUSION

We introduced a novel representation learning algorithm for natural language modelling that infers discrete, relational representations which allow for compositionality. Experiments show our model learns representations encoding high-level semantics of natural sentences, thereby adding some novel layers of interpretability to large, pretrained language models.

---

[4]Note, in particular, that BART (Lewis et al., 2020) has 400M parameter, whereas HSN has 250M.

## 8 REPRODUCIBILITY STATEMENT

We provide source code to reproduce our results as supplementary material. The README.rst file within it contains instructions to install and run the corresponding libraries. The synthetic datasets of section 4 are also provided within the source code file, in the data directory. All other datasets we used are available online and are automatically downloaded by our training scripts.

We additionally provide explicit derivation and/or details for all mathematical expression within the main text in the Appendix. Details on hyper-parameter selection and training can also be found in the Appendix.

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

# APPENDIX

## A    PSEUDO-SELF ATTENTION MECHANISM REVISITED

The attention mechanism of the original Transformers (Vaswani et al., 2017) is defined as

$$\text{Attention}(\mathbf{Q}, \mathbf{K}, \mathbf{V}) = \text{softmax}\left(D^{-\frac{1}{2}} \mathbf{Q} \cdot \mathbf{K}^T\right) \mathbf{V}, \tag{13}$$

where $\mathbf{Q}$, $\mathbf{K}$ and $\mathbf{V} \in \mathbb{R}^{T \times D}$ are sets of queries, keys and values, respectively, given by a sequence of $T$, $D$-dimensional vectors, packed into matrices. In practice, these queries, keys and values are projected many times with different learnable, linear maps. The Attention operation (Eq. 13) is performed on these different projections in parallel, whose outputs are then concatenated and projected once more with a final, linear map. The complete operation is known as Multi-head Attention (Vaswani et al., 2017), and we use this notation in Fig. 1 of the main text.

Now, the question is how to condition GPT-2 on the schema $\mathbf{e}_{j_1:j_L}$. Given a sequence of input representations $\mathbf{u}_{1:T}$, the *self*-attention mechanism in GPT-2 is obtained by choosing $\mathbf{Q} = \mathbf{u}_{1:T} \cdot \mathbf{W}_Q$, $\mathbf{K} = \mathbf{u}_{1:T} \cdot \mathbf{W}_K$ and $\mathbf{V} = \mathbf{u}_{1:T} \cdot \mathbf{W}_V$, all in $\mathbb{R}^{T \times D}$, with $\mathbf{W}_Q, \mathbf{W}_K$ and $\mathbf{W}_V \in \mathbb{R}^{D \times D}$ pretrained matrices. We leverage a pseudo-self attention (PSA) mechanism (Ziegler et al., 2019) that augments the key and value matrices in their first $L$ rows, with projections of $\mathbf{e}_{j_1:j_L}$ so that

$$\tilde{\mathbf{K}} = \begin{pmatrix} \mathbf{e}_{j_1:j_L} \cdot \mathbf{W}_K^e + \mathbf{p}_{\text{enc}} \\ \mathbf{K} \end{pmatrix}, \ \tilde{\mathbf{V}} = \begin{pmatrix} \mathbf{e}_{j_1:j_L} \cdot \mathbf{W}_V^e + \mathbf{p}_{\text{enc}} \\ \mathbf{V} \end{pmatrix} \in \mathbb{R}^{(L+T) \times D}, \tag{14}$$

where $\mathbf{p}_{\text{enc}}$ is a positional encoding, just as the one used in the original Transformer implementation (Vaswani et al., 2017). The latter informs GPT-2 about the ordering of the symbols in the schema, as selected by the random walk process. PSA is then simply given by Eq. 13 with the keys and values replaced with the augmented ones, $\tilde{\mathbf{K}}$ and $\tilde{\mathbf{V}}$. The $\mathbf{W}_K^e, \mathbf{W}_V^e$ here are randomly initialized, learnable parameters mapping the schemata onto the decoder self-attention, $D$-dimensional space, and we have as many of them as layers in GPT-2. Therefore this mechanism allows GPT-2 to attend to the projected schema at each of its layers, with a minimal addition of untrained parameters (Ziegler et al., 2019).

## B    TRAINING OBJECTIVE

The Evidence Lower Bound (ELBO) of the Hidden Schema Network model reads

$$\mathcal{L}[\theta, \phi] = \frac{1}{N} \sum_{n=1}^{N} \mathbb{E}_{q_\phi(\mathbf{z}_{1:L}|\mathbf{x}_{1:T}^{(n)}, \mathbf{A}) q_\phi(\mathbf{A})} \log p_\theta(\mathbf{x}_{1:T}^{(n)}|\mathbf{z}_{1:L})$$

$$- \mathbb{E}_{q_\phi(\mathbf{A})} \text{KL}\left[q_\phi(\mathbf{z}_{1:L}|\mathbf{x}_{1:T}^{(n)}, \mathbf{A}); p(\mathbf{z}_{1:L}|\mathbf{A})\right] - \text{KL}[q_\phi(\mathbf{A}); p(\mathbf{A})], \tag{15}$$

where $\text{KL}[\cdot]$ denotes the Kullback-Leibler (KL) divergence.

Note that this is *not* the training objective of the main text. There we maximize the ELBO together with the mutual information between sentences and schemata. We give details about this modified objective in subsection B.3 below. Before getting into that, let us first calculate the explicit expressions for the two divergences above.

### B.1    KULLBACK-LEIBLER BETWEEN RANDOM WALKS

For notational convenience we will not write the explicit dependence on the graph $\mathbf{A}$ in what follows. Using the explicit product form of the probabilities over walks leads to

$$\text{KL}[q_\phi(\mathbf{z}_{1:T}|\mathbf{x}_{1:T}^{(n)}); p(\mathbf{z}_{1:T})] = \sum_{i=2}^{L} \mathbb{E}_{\hat{q}_\phi(\mathbf{z}_{i-1}|\mathbf{x}_{1:T}^{(n)}) q_\phi(\mathbf{z}_i|\mathbf{z}_{i-1}\mathbf{x}_{1:T}^{(n)})} \log \frac{q_\phi(\mathbf{z}_i|\mathbf{z}_{i-1}, \mathbf{x}_{1:T}^{(n)})}{p(\mathbf{z}_i|\mathbf{z}_{i-1})}$$

$$+ \text{KL}[q_\phi(\mathbf{z}_1); p(\mathbf{z}_1)], \tag{16}$$

| Graph $\mathcal{G}^*$ | Model | NLL | KL $-$ z | KL $-$ $\mathcal{G}$ | AUC | $|\mathcal{G}^* - \mathcal{G}|_F$ | $|\mathcal{G}^\tau - \mathcal{G}|_F$ | N. edges($\mathcal{G}$) |
|---|---|---|---|---|---|---|---|---|
| | LSTM | **53.07**$\pm$ **0.01** | – | – | – | – | – | – |
| Barabasi | HS (0.1) | $53.08 \pm 0.01$ | $0.10 \pm 0.06$ | $9 \pm 1$ | $0.977 \pm 0.003$ | $17 \pm 2$ | $27 \pm 1$ | $1090 \pm 143$ |
| | HS (0.2) | $53.07 \pm 0.02$ | $0.09 \pm 0.06$ | $4.8 \pm 0.5$ | $0.989 \pm 0.001$ | $17 \pm 2$ | $26 \pm 1$ | $1360 \pm 104$ |
| | LSTM | **48.24** $\pm$ **0.02** | – | – | – | – | – | – |
| Erdos | HS (0.5) | $50.9 \pm 0.8$ | $1.2 \pm 0.3$ | $4 \pm 6$ | $0.95 \pm 0.06$ | $34.8 \pm 0.9$ | $40 \pm 5$ | $2812 \pm 344$ |
| | HS (0.6) | $50.4 \pm 0.6$ | $1.3 \pm 0.1$ | $1 \pm 2$ | $0.94 \pm 0.06$ | $36.8 \pm 0.8$ | $44 \pm 2$ | $3131 \pm 156$ |

Table 5: Inference on ground-truth random graphs. Here we use the notation HS($p$) to denote Hidden Schema Network models with prior graph distributions whose edge probability is set to $p$.

where $\hat{q}_\phi(\mathbf{z}_i|\mathbf{x}_{1:T}^{(n)})$ is the aggregated probability over all walks until step $i$. Since the random walks are Markovian, $\hat{q}$ can be explicitly written as

$$\hat{q}_\phi(\mathbf{z}_i|\mathbf{x}_{1:T}^{(n)}) = \prod_{1 \leq j < i} \mathbf{Q}^{[j]}(\mathbf{x}_{1:T}^{(n)}, \phi) \cdot \boldsymbol{\rho}(\mathbf{x}_{1:T}^{(n)}, \phi), \tag{17}$$

where the (posterior) class probabilities over the walks' starting points $\boldsymbol{\rho}$, and the transition matrices $\mathbf{Q}^{[i]}$ are defined in Eqs. 10 and 11 of the main text. Using the definitions in Eqs. 4 and 9 we can write the argument of the expectation value in Eq. 16 above as

$$\log \frac{q_\phi(\mathbf{z}_i|\mathbf{z}_{i-1}, \mathbf{x}_{1:T}^{(n)})}{p(\mathbf{z}_i|\mathbf{z}_{i-1})} = \sum_{k,j} z_i^k z_{i-1}^j \log \frac{Q_{k,j}^{[i-1]}(\mathbf{x}_{1:T}^{(n)}, \phi)}{P_{k,j}}, \tag{18}$$

which means we only need to compute the expectation of the product $z_i^k z_{i-1}^j$. This one can easily be shown to be

$$\mathbb{E}_{\hat{q}_\phi(\mathbf{z}_{i-1}|\mathbf{x}_{1:T}^{(n)})q_\phi(\mathbf{z}_i|\mathbf{z}_{i-1}\mathbf{x}_{1:T}^{(n)})}\left[z_i^k z_{i-1}^j\right] = Q_{k,j}^{[i-1]}(\mathbf{x}_{1:T}^{(n)}, \phi)\, \hat{\rho}_j^{[i-1]}(\mathbf{x}_{1:T}^{(n)}, \phi), \tag{19}$$

where $\hat{\rho}_j^{[i]}(\mathbf{x}_{1:T}^{(n)}, \phi)$ is the $j$th class probability of $\hat{q}_\phi(\mathbf{z}_i|\mathbf{x}_{1:T}^{(n)})$, defined in Eq. 17.

Finally, the second KL term in Eq. 16 can be directly evaluated

$$\text{KL}[q_\phi(\mathbf{z}_1); p(\mathbf{z}_1)] = \sum_{j=1}^{K} \rho_j(\mathbf{x}_{1:T}^{(n)}, \phi) \log \frac{\rho_j(\mathbf{x}_{1:T}^{(n)}, \phi)}{\rho_j}, \tag{20}$$

where $\rho_j(\mathbf{x}_{1:T}^{(n)}, \phi)$ and $\rho_j$ are, respectively, the posterior and prior class probabilities for the random walks' starting points.

| Method | PTB | YAHOO | YELP |
|---|---|---|---|
| GPT2 [one epoch] | 24.23 | 22.00 | 23.40 |
| GPT2 [fine-tuned] | 19.14 | 20.64 | 19.77 |
| iVAE$_{\text{MI}}$ | 53.44 | 47.93 | 36.88 |
| Optimus | 23.58 | 22.34 | 21.99 |
| HSN $(100, 20)$ | 17.61 | 19.68 | 18.99 |
| HSN $(100, 5)$ | 17.69 | 19.84 | 19.00 |
| HSN $(50, 20)$ | **16.88** | **19.59** | 19.01 |
| HSN $(50, 5)$ | 17.41 | 20.06 | **18.95** |
| HSN $(100, 20)$ | $17.6 \pm 0.1$ | $20.1 \pm 0.2$ | $19.12 \pm 0.08$ |
| HSN $(100, 5)$ | $17.79 \pm 0.09$ | $20.0 \pm 0.1$ | $19.07 \pm 0.08$ |
| HSN $(50, 20)$ | $17.0 \pm 0.1$ | $19.8 \pm 0.2$ | $19.3 \pm 0.5$ |
| HSN $(50, 5)$ | $17.44 \pm 0.07$ | $20.09 \pm 0.05$ | $18.9 \pm 0.1$ |

Table 6: Perplexity per word (lower is better). The last four (4) rows show the mean and standard deviation obtained after training five (5) HSN.

| Model | PTB | | YAHOO | | YELP | |
|---|---|---|---|---|---|---|
| | $KL - \mathbf{z}$ | $KL - \mathcal{G}$ | $KL - \mathbf{z}$ | $KL - \mathcal{G}$ | $KL - \mathbf{z}$ | $KL - \mathcal{G}$ |
| HS [$L = 20$] | $2.0 \pm 0.3$ | $0.004 \pm 0.002$ | $1.3 \pm 0.6$ | $0.005 \pm 0.003$ | $1.9 \pm 0.7$ | $0.011 \pm 0.002$ |
| HS [$L = 5$] | $1.33 \pm 0.05$ | $0.0013 \pm 0.0009$ | $1.54 \pm 0.05$ | $0.0006 \pm 0.0006$ | $1.50 \pm 0.09$ | $0.002 \pm 0.002$ |

Table 7: Kullback-leibler divergence for 100-symbol HSN models (trained 5 times) in all datasets

Putting all together we write

$$
\begin{aligned}
\text{KL}[q_\phi(\mathbf{z}_{1:T}|\mathbf{x}_{1:T}^{(n)}); p(\mathbf{z}_{1:T})] = &\sum_{i=2}^{L} \sum_{k,j=1}^{K} Q_{k,j}^{[i-1]}(\mathbf{x}_{1:T}^{(n)}, \phi)\, \hat{\rho}_j^{[i-1]}(\mathbf{x}_{1:T}^{(n)}, \phi) \log \frac{Q_{k,j}^{[i-1]}(\mathbf{x}_{1:T}^{(n)}, \phi)}{P_{k,j}} \\
&+ \sum_{j=1}^{K} \rho_j(\mathbf{x}_{1:T}^{(n)}, \phi) \log \frac{\rho_j(\mathbf{x}_{1:T}^{(n)}, \phi)}{\rho_j}
\end{aligned}
\tag{21}
$$

## B.2 KULLBACK-LEIBLER BETWEEN RANDOM GRAPH MODELS

Since both prior and posterior graph models treat each edge in $\mathcal{G}$ as a Bernoulli random variable, we can write directly

$$
\begin{aligned}
\text{KL}[q(\mathbf{A})|p(\mathbf{A})] = \sum_{ij} \Bigg\{ &p_\phi(\mathbf{e}_i, \mathbf{e}_j) \log \left( \frac{p_\phi(\mathbf{e}_i, \mathbf{e}_j)}{p} \right) \\
&+ (1 - p_\phi(\mathbf{e}_i, \mathbf{e}_j)) \log \left( \frac{1 - p_\phi(\mathbf{e}_i, \mathbf{e}_j)}{1 - p} \right) \Bigg\},
\end{aligned}
\tag{22}
$$

where $p_\phi(\mathbf{e}_i, \mathbf{e}_j)$ is the posterior link probability, which is conditioned on the symbols connected by the link, and $p$ is the global prior probability over all links, as defined in Eq. 3 of the main text.

## B.3 MAXIMIZING MUTUAL INFORMATION

We would like to maximize the mutual information between the word sequences in our dataset and the schema representations. We have argued that the training objective in the main text already includes such a mutual information term. To see this is indeed the case we need to workout some identities.

Let us, for simplicity of notation, consider two discrete variables $\mathbf{z}$ and $\mathbf{x}$, the last of which follows an unknown distribution $p_\mathcal{D}(\mathbf{x})$. What follow are identities

$$
\begin{aligned}
-\mathbb{E}_{p_\mathcal{D}(\mathbf{x})} \text{KL}[q(\mathbf{z}|\mathbf{x}); p(\mathbf{z})] &= \mathbb{E}_{p_\mathcal{D}(\mathbf{x})} \mathbb{E}_{q(\mathbf{z}|\mathbf{x})} \Big\{ \log p(\mathbf{z}) - \log(\mathbf{z}|\mathbf{x}) \Big\} \\
&= H_q(\mathbf{z}|\mathbf{x}) + \sum_{\mathbf{x}} p_\mathcal{D}(\mathbf{x}) \sum_{\mathbf{z}} q(\mathbf{z}|\mathbf{x}) \Big\{ \log p(\mathbf{z}) + \log q^*(\mathbf{z}) - \log q^*(\mathbf{z}) \Big\} \\
&= H_q(\mathbf{z}|\mathbf{x}) - H_{q^*}(\mathbf{z}) + \sum_{\mathbf{z}} q^*(\mathbf{z}) \Big\{ \log p(\mathbf{z}) - \log q^*(\mathbf{z}) \Big\} \\
&= -I(\mathbf{z}; \mathbf{x}) - \mathbb{E}_{q^*(\mathbf{z})} \Big\{ \log \frac{q^*(\mathbf{z})}{p(\mathbf{z})} \Big\} \\
&= -I(\mathbf{z}; \mathbf{x}) - \text{KL}[q^*(\mathbf{z}); p(\mathbf{z})],
\end{aligned}
\tag{23}
$$

where

$$
H_q(\mathbf{z}|\mathbf{x}) = -\sum_{\mathbf{x}} p_\mathcal{D}(\mathbf{x}) \sum_{\mathbf{z}} q(\mathbf{z}|\mathbf{x}) \log q(\mathbf{z}|\mathbf{x}),
\tag{24}
$$

is the conditional entropy with respect to distribution $q$ (see e.g. page 17 in (Cover & Thomas, 1991)) and

| | Dataset | n. edges | $\mathcal{D}$ | $l$ | $\mathcal{C}$ | $\mathcal{CC}$ | largest $\mathcal{CC}$ |
|---|---|---|---|---|---|---|---|
| HSN(50, 5) | PTB | $694.26 \pm 9.47$ | $2.00 \pm 0.00$ | $1.43 \pm 0.01$ | $0.83 \pm 0.01$ | $1.00 \pm 0.00$ | $50.00 \pm 0.00$ |
| | YAHOO | $892.67 \pm 8.22$ | $2.00 \pm 0.00$ | $1.24 \pm 0.01$ | $0.84 \pm 0.00$ | $2.00 \pm 0.04$ | $49.00 \pm 0.04$ |
| | YELP | $891.06 \pm 6.50$ | $2.73 \pm 0.46$ | $1.24 \pm 0.03$ | $0.84 \pm 0.01$ | $2.24 \pm 0.85$ | $48.76 \pm 0.85$ |
| | Random | $611.69 \pm 17.61$ | $2.00 \pm 0.00$ | $1.50 \pm 0.01$ | $0.50 \pm 0.02$ | $1.00 \pm 0.00$ | $50.00 \pm 0.00$ |
| HSN(50, 20) | PTB | $764.28 \pm 7.88$ | $2.83 \pm 0.38$ | $1.27 \pm 0.03$ | $0.82 \pm 0.01$ | $4.71 \pm 0.77$ | $46.29 \pm 0.77$ |
| | YAHOO | $356.35 \pm 7.76$ | $3.17 \pm 0.37$ | $1.57 \pm 0.05$ | $0.58 \pm 0.02$ | $12.04 \pm 1.49$ | $38.96 \pm 1.49$ |
| | YELP | $259.42 \pm 5.47$ | $2.68 \pm 0.48$ | $1.42 \pm 0.03$ | $0.48 \pm 0.01$ | $20.77 \pm 0.68$ | $30.23 \pm 0.68$ |
| | Random | $611.69 \pm 17.61$ | $2.00 \pm 0.00$ | $1.50 \pm 0.01$ | $0.50 \pm 0.02$ | $1.00 \pm 0.00$ | $50.00 \pm 0.00$ |
| HLN(100, 5) | PTB | $1198.18 \pm 16.44$ | $2.56 \pm 0.50$ | $1.76 \pm 0.01$ | $0.83 \pm 0.01$ | $1.19 \pm 0.40$ | $99.81 \pm 0.40$ |
| | YAHOO | $1239.21 \pm 12.19$ | $3.15 \pm 0.38$ | $1.42 \pm 0.03$ | $0.51 \pm 0.01$ | $35.93 \pm 1.41$ | $65.07 \pm 1.41$ |
| | YELP | $1295.68 \pm 12.93$ | $3.36 \pm 0.48$ | $1.55 \pm 0.03$ | $0.57 \pm 0.01$ | $27.38 \pm 1.69$ | $73.62 \pm 1.69$ |
| | Random | $2474.92 \pm 36.58$ | $2.00 \pm 0.00$ | $1.50 \pm 0.01$ | $0.50 \pm 0.01$ | $1.00 \pm 0.00$ | $100.00 \pm 0.00$ |
| HLN(100, 20) | PTB | $892.53 \pm 10.04$ | $3.04 \pm 0.24$ | $1.41 \pm 0.04$ | $0.45 \pm 0.01$ | $46.04 \pm 1.53$ | $54.96 \pm 1.54$ |
| | YAHOO | $261.13 \pm 7.14$ | $2.18 \pm 0.38$ | $1.95 \pm 0.00$ | $0.91 \pm 0.01$ | $1.01 \pm 0.10$ | $99.99 \pm 0.10$ |
| | YELP | $515.84 \pm 10.09$ | $3.68 \pm 0.48$ | $1.79 \pm 0.06$ | $0.38 \pm 0.02$ | $45.27 \pm 2.58$ | $55.67 \pm 2.58$ |
| | Random | $2474.92 \pm 36.58$ | $2.00 \pm 0.00$ | $1.50 \pm 0.01$ | $0.50 \pm 0.01$ | $1.00 \pm 0.00$ | $100.00 \pm 0.00$ |

Table 8: Statistic of inferred graphs for all datasets

$$H_{q^*}(\mathbf{z}) = -\sum_{\mathbf{z}} q^*(\mathbf{z}) \log q^*(\mathbf{z}), \tag{25}$$

is the entropy of distribution $q^*(\mathbf{z})$, which we define as the marginal (data-aggregated) distribution

$$q^*(\mathbf{z}) = \sum_{\mathbf{x}} p_{\mathcal{D}}(\mathbf{x}) q(\mathbf{z}|\mathbf{x}). \tag{26}$$

Finally, we used the definition of mutual information

$$I(\mathbf{x}; \mathbf{z}) = H_{q^*}(\mathbf{z}) - H_q(\mathbf{z}|\mathbf{x}). \tag{27}$$

See e.g. page 20 in (Cover & Thomas, 1991).

It follows from Eq. 23 that maximizing the ELBO (Eq. 15), together with the mutual information between word sequences and schemata, simply amounts to replacing the KL between the approximate posterior and prior random walk distributions, with the KL between the *aggregated posterior* and prior random walk distributions. To wit

$$-\frac{1}{N} \sum_{n=1}^{N} \mathbb{E}_{q_\phi(\mathbf{A})} KL\Big[ q_\phi(\mathbf{z}_{1:L}|\mathbf{x}_{1:T}^{(n)}, \mathbf{A}); p(\mathbf{z}_{1:T}|\mathbf{A}) \Big] + I(\mathbf{z}_{1:L}; \mathbf{x}_{1:T}|\mathbf{A}) =$$
$$-\mathbb{E}_{q_\phi(\mathbf{A})} KL\Big[ q_\phi^*(\mathbf{z}_{1:L}|\mathbf{A}); p(\mathbf{z}_{1:T}|\mathbf{A}) \Big], \tag{28}$$

where we introduced the aggregated posterior over random walks wrt the word sequence

$$q_\phi^*(\mathbf{z}_{1:L}) = \mathbb{E}_{p(\mathbf{x}_{1:T})} \Big[ q_\phi(\mathbf{z}_{1:L}|\mathbf{x}_{1:T}) \Big] \approx \frac{1}{N} \sum_{n=1}^{N} q_\phi(\mathbf{z}_{1:L}|\mathbf{x}_{1:T}^{(n)}). \tag{29}$$

In practice we approximate this quantity with

$$q_\phi^*(\mathbf{z}_{1:L}) \approx q_\phi^*(\mathbf{z}_1) \prod_{i=2}^{L} q_\phi^*(\mathbf{z}_i|\mathbf{z}_{i-1}, \mathbf{A}), \tag{30}$$

where $q_\phi^*(\mathbf{z}_1)$ is a categorical distribution whose class probabilities $\rho_j^*(\phi)$ are the average of those from our approximate posterior (Eq. 10 in the main text)

$$\rho_j^*(\phi) = \frac{1}{N} \sum_{n=1}^{N} \rho_j(\mathbf{x}_{1:T}^{(n)}, \phi), \tag{31}$$

and the transition probabilities $q_\phi^*(\mathbf{z}_i|\mathbf{z}_{i-1}, \mathbf{A})$ have transition probability matrices

$$Q_{k,j}^{*\,[i]}(\mathbf{A}, \phi) = \frac{1}{N} \sum_{n=1}^{N} Q_{k,j}^{[i]}(\mathbf{x}_{1:T}^{(n)}, \mathbf{A}, \phi). \tag{32}$$

## B.4 MEAN-FIELD SOLUTION

Instead of modeling the posterior over random walks with Eq. 9 of the main text, we could consider a mean-field decomposition along the time component, by ignoring the dependency on the graph $\mathcal{G}$

$$q_\phi(\mathbf{z}_{1:L}|\mathbf{x}_{1:T}) = \prod_{i=1}^{L} q_\phi(\mathbf{z}_i|\mathbf{x}_{1:T}), \tag{33}$$

where at each step of the walk we have a step-dependent categorical distribution

$$q_\phi(\mathbf{z}_i|\mathbf{x}_{1:T}) = \prod_{j=1}^{K} \left(\rho_j^{[i]}(\mathbf{x}_{1:T}, \phi)\right)^{z_i^j}, \tag{34}$$

whose class probabilities live in the $K$-simplex. We could model the latter via

$$\boldsymbol{\rho}^{[1]}, \ldots, \boldsymbol{\rho}^{[L]} = \text{softmax}(\mathbf{h}_1^{\text{enc}}, \ldots, \mathbf{h}_L^{\text{enc}}) \tag{35}$$

where $\mathbf{h}_1^{\text{enc}}, \ldots, \mathbf{h}_L^{\text{enc}}$ are the outputs of our encoder neural network model, shown in Figure 1 of the main text.

Replacing the mean-field approximation of 33 into 15 yields

$$
\begin{aligned}
KL[q_\phi(\mathbf{z}_{1:T}|\mathbf{x}_{1:T}^{(n)}); p(\mathbf{z}_{1:T}|\mathbf{A})] = \sum_{i=2}^{L} \Big\{ &\mathbb{E}_{q_\phi(\mathbf{z}_i|\mathbf{x}_{1:T}^{(n)})} \log q_\phi(\mathbf{z}_i|\mathbf{x}_{1:T}^{(n)}) \\
&- \mathbb{E}_{q_\phi(\mathbf{z}_i|\mathbf{x}_{1:T}^{(n)})q_\phi(\mathbf{z}_{i-1}|\mathbf{x}_{1:T}^{(n)})} \log p(\mathbf{z}_i|\mathbf{z}_{i-1}) \Big\} + KL[q_\phi(\mathbf{z}_1); p(\mathbf{z}_1)], \\
= \sum_{i=1}^{L} \sum_{j}^{K} &\rho_j^{[i]}(\mathbf{x}_{1:T}, \phi) \log \frac{\rho_j^{[i]}(\mathbf{x}_{1:T}, \phi)}{\rho_j} \\
- \sum_{i=2}^{L} \sum_{k,j}^{K} &\mathbb{E}_{q_\phi(\mathbf{z}_i|\mathbf{x}_{1:T}^{(n)})q_\phi(\mathbf{z}_{i-1}|\mathbf{x}_{1:T}^{(n)})} \left[z_i^k z_{i-1}^j\right] \log P_{k,j} \\
= \sum_{i=1}^{L} \sum_{j}^{K} &\rho_j^{[i]}(\mathbf{x}_{1:T}, \phi) \log \frac{\rho_j^{[i]}(\mathbf{x}_{1:T}, \phi)}{\rho_j} \tag{36} \\
- \sum_{i=2}^{L} \sum_{k,j}^{K} &\rho_k^{[i]}(\mathbf{x}_{1:T}, \phi) \rho_j^{[i-1]}(\mathbf{x}_{1:T}, \phi) \log P_{k,j}. \tag{37}
\end{aligned}
$$

## B.5 FULLY CONNECTED GRAPH

We can replace the adjacency matrix $\mathbf{A}$ in the definition of the transition probability matrix of our posterior $\mathbf{Q}(\mathbf{x}_{1:T}, \mathbf{A}, \phi)$, with that of a fully connected graph. The aggregated posterior over all walks up to step $i$ (Eq. 17 above) reduces in this case to

$$
\begin{aligned}
\hat{\rho}_k^{[i]}(\mathbf{x}_{1:T}, \phi) &= \sum_{j}^{K} \left(\frac{f_k^{[i-1]}(\mathbf{x}_{1:T}, \phi) A_{k,j}}{\sum_m f_m^{[i-1]}(\mathbf{x}_{1:T}, \phi) A_{m,j}}\right) \hat{\rho}_j^{[i-i]}(\mathbf{x}_{1:T}, \phi) \\
&= \left(\frac{f_k^{[i-1]}(\mathbf{x}_{1:T}, \phi)}{\sum_m f_m^{[i-1]}(\mathbf{x}_{1:T}, \phi)}\right) \left(\sum_{j}^{K} \hat{\rho}_j^{[i-i]}(\mathbf{x}_{1:T}, \phi)\right) = \frac{f_k^{[i-1]}(\mathbf{x}_{1:T}, \phi)}{\sum_m f_m^{[i-1]}(\mathbf{x}_{1:T}, \phi)} \tag{38}
\end{aligned}
$$

which is equivalent to that of the mean-field approximation of section B.4 with $\hat{\rho}_k^{[i]} = \rho_k^{[i]}$.

## C  HIDDEN SCHEMA NETWORKS ALGORITHM

---

**Algorithm 1:** HSN Training $(\phi, \psi)$

---

**foreach** *minibatch* $\mathbf{x}_{1:T} \sim p(\mathcal{D})$ **do**

    **(1) Sample schema network from posterior graph model:**

$$\mathbf{A} \quad \sim \quad q_\phi(\mathbf{A}),$$

    **(2) Compute parameters of posterior random walk model:**

$$
\begin{aligned}
\mathbf{h}_1^{\mathrm{enc}}, \mathbf{h}_2^{\mathrm{enc}} \ldots, \mathbf{h}_L^{\mathrm{enc}} &= \mathbf{h}_\phi^{\mathrm{enc}}(\mathbf{x}_{1:T}), \\
\boldsymbol{\rho}(\phi) &= \mathrm{softmax}(\mathbf{h}_1^{\mathrm{enc}}), \\
Q_{k,j}^{[i]}(\phi) &= \frac{f_k^{[i]}(\phi)\, A_{kj}}{\sum_m f_m^{[i]}(\phi)\, A_{mj}}, \quad \text{with } \mathbf{f}^{[1]}, \ldots, \mathbf{f}^{[L-1]} = \exp(\mathbf{h}_{2:L}^{\mathrm{enc}})
\end{aligned}
$$

    **(3) Compute parameters of prior random walk model:**

$$P_{k,j} = \frac{f_k\, A_{kj}}{\sum_{i=1}^{K} f_i\, A_{ij}}$$

    **(4) Sample random walks from posterior distribution:**

$$\mathbf{z}_{1:L} \sim q_\phi(\mathbf{z}_{1:L} | \mathbf{x}_{1:T}, \mathbf{A})$$

    **(5) Decode sentence:**

    **for** $i = 0$ **to** $T-1$ **do**

$$\mathbf{x}_i \sim p_\theta(\mathbf{x}_i | \mathbf{x}_{<i}, \mathbf{e}_{j_1:j_L}), \quad \boldsymbol{\pi}_i = \mathrm{softmax}(\mathbf{W} \cdot \mathbf{h}_\theta^{\mathrm{dec}}(\mathbf{x}_{<i}, \mathbf{e}_{j_1:j_L}))$$

    **end**

    **(6) Compute loss and back-propagate:**

$$
\begin{aligned}
\mathcal{L}[\theta, \phi] = \frac{1}{N} \sum_{n=1}^{N} & \mathbb{E}_{q_\phi(\mathbf{z}_{1:L}|\mathbf{x}_{1:T}^{(n)}, \mathbf{A})q_\phi(\mathbf{A})} \log p_\theta(\mathbf{x}_{1:T}^{(n)} | \mathbf{z}_{1:L}) \\
& - \mathbb{E}_{q_\phi(\mathbf{A})} \mathrm{KL}\Big[ q_\phi^*(\mathbf{z}_{1:L}|\mathbf{A}); p(\mathbf{z}_{1:L}|\mathbf{A}) \Big] - \mathrm{KL}[q_\phi(\mathbf{A}); p(\mathbf{A})]
\end{aligned}
$$

**end**

---

## D  ON SYNTHETIC DATASET EXPERIMENTS

In this section we give additional details of and results from our proof-of-concept experiments.

### D.1  SYNTHETIC LANGUAGE MODEL

We generate our synthetic dataset as follows: first, we sample a single, fixed graph $\mathcal{G}^*$ with $K$ nodes from a predefined random graph model. Second, we define a set of random tokens $\mathcal{V}$, of size $V$, to be our vocabulary. We create each token as a random 3-tuple from the Latin alphabet, and choose to have at least one order of magnitude more tokens than nodes in $G$ (that is, $V \gg K$). Third, we assign a random bag of tokens to each node in $\mathcal{G}^*$. These random bags can simply be understood as probability distributions over $\mathcal{V}$, and can be represented as $V$-dimensional vectors whose components live on the simplex. Note in particular that, by construction, tokens can be shared among the different nodes of $\mathcal{G}^*$. Finally, let us identify the $K$ random bags with the $K$ symbols $\{\mathbf{e}_1, \mathbf{e}_2, \ldots, \mathbf{e}_K\}$ of the synthetic language model.

To generate synthetic sentences we sample uniform, $L$-step random walks on $\mathcal{G}^*$, whose transition matrix is given by Eq. 4 in the main text, with $\mathbf{f} = \mathbb{I}$. Having obtained a set of random walks on $\mathcal{G}^*$, we sample one random token from each of the symbols (i.e. from each random bag) along the walks.

## D.2 EXPERIMENTAL SETTINGS

Here we give additional details for reproducibility

**Datasets**

- Following the procedure above we generated two datasets from two random graphs with different topologies. One sampled from the Barabási-Albert model (Barabási & Albert, 1999), the other from the Erdös-Rényi model (Erdös & Rényi, 1959). We generate these graphs using NetworkX, a Python language software package for network structures (Hagberg et al., 2008). Specifically, we generate Barabási-Albert graphs by attaching 3 edges from each new node to old ones, and Erdös-Rényi graphs with an edge probability of 0.5. We set both graphs to have $K = 100$ symbols.
- We define each random bag of tokens in $\mathcal{G}^*$ to have two tokens only (each with equal probability).
- We use a vocabulary of 1000 random tokens.
- Once the graph is fixed, we set the token sequence length to $L = 10$ ($L = 11$) for the Erdös (Barabási) datasets and generate a total of $N = 100000$ token sequences from each random graph.

**Hidden Schema Network** (HSN) **settings**

- We train randomly initialized embeddings of dimension 256, one for each token. We sample these from a normal distribution with zero mean and a standard deviation of 0.01.
- The posterior graph model is defined via a single feed-forward neural network with 256 hidden units.
- The prior graph model has the edge probability $p$ as hyperparameter. We crossvalidate it from the set $p = \{0.1, 0.2, 0.5, 0.6, 0.8\}$ and found that HSN could fit the Barabási dataset only with small values $\{0.1, 0.2\}$. HSN could fit the Erdös dataset with larger values $\{0.5, 0.6\}$
- The posterior random walk model is defined by replacing BERT with a 2-block Transformer encoder (Vaswani et al., 2017), each with 2 heads, 256 hidden units and dropout probability of 0.2.
- The prior random walk model was set to a uniform random walk.

**Training details**

- We use a batch size of 256 and train with Adam (Kingma & Ba, 2014), with a learning rate of 0.0001, in all experiments.
- To sample both graph and random walk posterior models with use the Gumbel-Softmax trick (Jang et al., 2016), with a constant temperature of 0.75
- We train the models for 200 epochs

## D.3 ADDITIONAL RESULTS

Table 5 displays the mean and standard deviation of some additional results on our proof-of-concept experiments. We trained ten models in total.

We first trained a simple LSTM Network to infer the correct symbol order in each random token sequence. We noticed that a network with 256 hidden units was enough to solve this task perfectly. Indeed, the negative log-likelihood (NLL) of these models corresponds to choosing the 2-token random bag sequence (i.e. the schema) that yields the correct token sequence without errors. The HSN performs equally well on the Barabási dataset, and slightly worst on the Erdös dataset. In fact, we have noticed the Erdös dataset proved to be more challenging to learn with the HSN in all regards.

See, for example, the AUC scores or the Frobenious norms of HSN in this dataset, as compared to the Barabási case. We think this might be due to the fact that Barabási graphs have more structure, simply because of their sparsity, which arguably make them easier to infer with our inductive bias.

Note also how increasing the prior edge probability $p$ affects the average number of edges of the inferred graphs.

# E    ON LANGUAGE MODELLING EXPERIMENTS

In this section we give additional details of and results from our language modelling and representation learning experiments.

## E.1    EXPERIMENTAL SETTINGS

Here we give additional details for reproducibility

**Datasets**

- We consider three widely used public datasets, namely the Penn Treebank (PTB) (Marcus et al., 1993), Yahoo and Yelp (Yang et al., 2017) corpora.

- PTB training set has a total of 38219 sentences. The average length of which is of about 22 words. The validation and test set have 5527 and 5462 sentences, respectively. The minimum (maximum) sentence length in PTB is of 2 (78) words.

- Yahoo training set has a total of 100000 sentences. The average length of which is of about 80 words. The validation and test sets have 10000 sentences each. The minimum (maximum) sentence length in Yahoo is of 21 (201) words. The Vocabulary size is of 200000 words.

- Yelp training set has a total of 100000 sentences. The average length of which is of about 97 words. The validation and test sets have 10000 sentences each. The minimum (maximum) sentence length in Yelp is of 21 (201) words. The Vocabulary size is of 90000 words.

**HSN settings**

- In all experiments we leveraged pretrained BERT and GPT-2 models, both with 12 layers, 768 hidden dimensions ($D$) and 12 attention heads. We used the public HuggingFace implementation of both these models (Wolf et al., 2020).

- The posterior graph model is set to a 2-layer feed forward network, each with hidden dimension 512.

- We crossvalidated the prior edge probability over the set of values $p = \{0.1, 0.2, 0.5, 0.6\}$ and found $p = 0.5$ (a maximum entropy prior) to yield the best results. All results we report correspond to this ($p = 0.5$) case.

- We also train an inhomogeneous random walk prior model by making $\boldsymbol{\rho}$ and the sequence of weights $\mathbf{f}^{[1]}, \mathbf{f}^{[2]}, \ldots, \mathbf{f}^{[L-1]}$ trainable. We initialized them by sampling from a normal distribution with zero mean and standard deviation of 0.01.

- We experimented with HSN of $K = \{50, 100\}$ symbols and random walks of length $L = \{5, 20\}$.

**Training details**

- We used a batch size of 32 and train with Adam (Kingma & Ba, 2014), with a learning rate of 0.00001, in all experiments.

- To sample both graph and random walk posterior models with used the Gumbel-Softmax trick (Jang et al., 2016), with a constant temperature of 1.0.

- We used a cyclical schedule to anneal both KL terms in our training objective from zero to one (Fu et al., 2019). When the annealing weight (usually called $\beta$ in the literature) is finite, we used a KL threshold scheme (Li et al., 2019), with a threshold value of 0.1.

- We trained the models for 100 epochs, although they usually needed about 60 epochs only to converge (in the NLL).
- We applied word dropout to the input of the decoder model with probability 0.3 in the following cases: (i) for all models trained on PTB; (ii) and all models with $L = 50$ trained on all datasets.

### E.2 ADDITIONAL RESULTS

Here we report results complementing the conclusions of the main text.

**Language modelling**. Table 6 displays our perplexity results on all datasets, just as in the main text. In the last four rows we additionally report the mean and standard deviation we obtained when repeating the experiments with the HSN model five times, with different initializations. The conclusion of the main text, viz. that our results outperform all baselines, remains unaltered, even within error bars. We additionally report in Table 7 the mean values of the KL for five 100-symbol HSN runs.

**Graph statistics**. We characterize the structure of $\mathcal{G}$ in terms of five statistics: (i) the diameter $\mathcal{D}$, which measures the maximum path length over all node pairs in $\mathcal{G}$; (ii) the average distance $l$, which instead measures the average shortest path length between all node pairs; (iii) the clustering coefficient $\mathcal{C}$, which represents the probability that two neighbors of a randomly chosen node are themselves neighbors; (iv) the number of connected components $\mathcal{CC}$; and (v) the degree distribution $P(k)$, which represents the probability that a randomly chosen node will have $k$ neighbors.

Table 8 reports the statistics of our inferred graphs for all datasets, and all model configurations.

We can see that increasing the random walk length from 5 to 20 increases the number of connected components of the graphs. As a consequence, subsets of word sequences are map onto smaller subgraphs, the larger of which is about 50 symbols. One could argue that, since longer random walk lengths imply a larger set of possible schema configurations, the number of symbols required to describe our three corpora can simply decrease. In other words, less symbols are needed by long schemata. Similarly, directly increasing the symbols number leads too to a larger number of connected components. Indeed, even the short schemata in Yelp and Yahoo do not use all available symbols to model the corpora.

**Representation learning**. We can get a graphical picture of the features we just discussed above in Figures 8–10 below. Very importantly, we see that the schema distribution is different for each category of each corpora in all model configurations. In other words, we do not observe any kind of mode collapse.

Finally, we have also explored "schema interpolations": given two schemata $e_{j_1:j_L}$ and $e_{m_1:m_L}$, we find the shortest path (of length $l$) on $\mathcal{G}$ connecting the end of $e_{j_1:j_L}$ with the beginning of $e_{m_1:m_L}$. Our interpolation steps are the schemata $\{e_{j_1+i:j_L+i} : \forall\ 0 \leq i \leq l + L$ along the path$\}$. Tables 10–12 show interpolations of random instances from all datasets. Note how the model successfully interpolate between categories in both Yelp and Yahoo.

| layer | KL(*good, bad*) | KL(*good, great*) | KL(*great, bad*) |
|-------|-----------------|-------------------|------------------|
| 1 | 0.807 | 0.336 | 1.227 |
| 5 | 0.738 | 0.177 | 1.245 |
| 12 | 0.635 | 0.224 | 0.957 |

Table 9: Kullback-Leibler divergence between the distributions of most attended symbols, when generating the tokens *good*, *bad* and *great*. Results are computed with HSN(100, 5) trained on Yelp. The KL values are computed for each head separately and then averaged.

## F    WHICH SYMBOLS DO WORDS ATTEND TO? A PRELIMINARY STUDY ON YELP REVIEWS

In this section we investigate how symbols are used by HSN when generating text. We do this by exploring the decoder attention matrix between the symbols and the generated tokens. Reading the attention wights, we can examine which symbols are most important for the generation of any given token, i.e. which symbols are attended to more strongly. A bit more in detail we select, for a given token in a given sentence, the symbol with the highest attention value. We can then compute the distribution of most attended symbols when generating that token for the complete dataset.

Thus, for a model trained on the Yelp dataset, we examine to which symbols does the decoder of HSN attend to, when processing the words *good*, *great* and *bad*. Figure 4 shows the most attended symbol distribution for layers 1 (first), 5 (middle), 12 (last), when averaging the attention matrices over all attention heads. Figures 5, 6, 7 show these distributions for each head separately. Note how, for a fixed token, the distribution of attention changes as one moves between heads and layers, albeit there are too some repeating patterns.

We can quantify these features by computing the Kullback-Leibler (KL) divergence between these distributions. The KL values are shown in Table 9.

Interestingly enough, the distribution of symbols that are attended to when processing the word *great* is closer to the distributions of symbols attended by the word *good*, than to the distributions of symbols attended by the word *bad*.

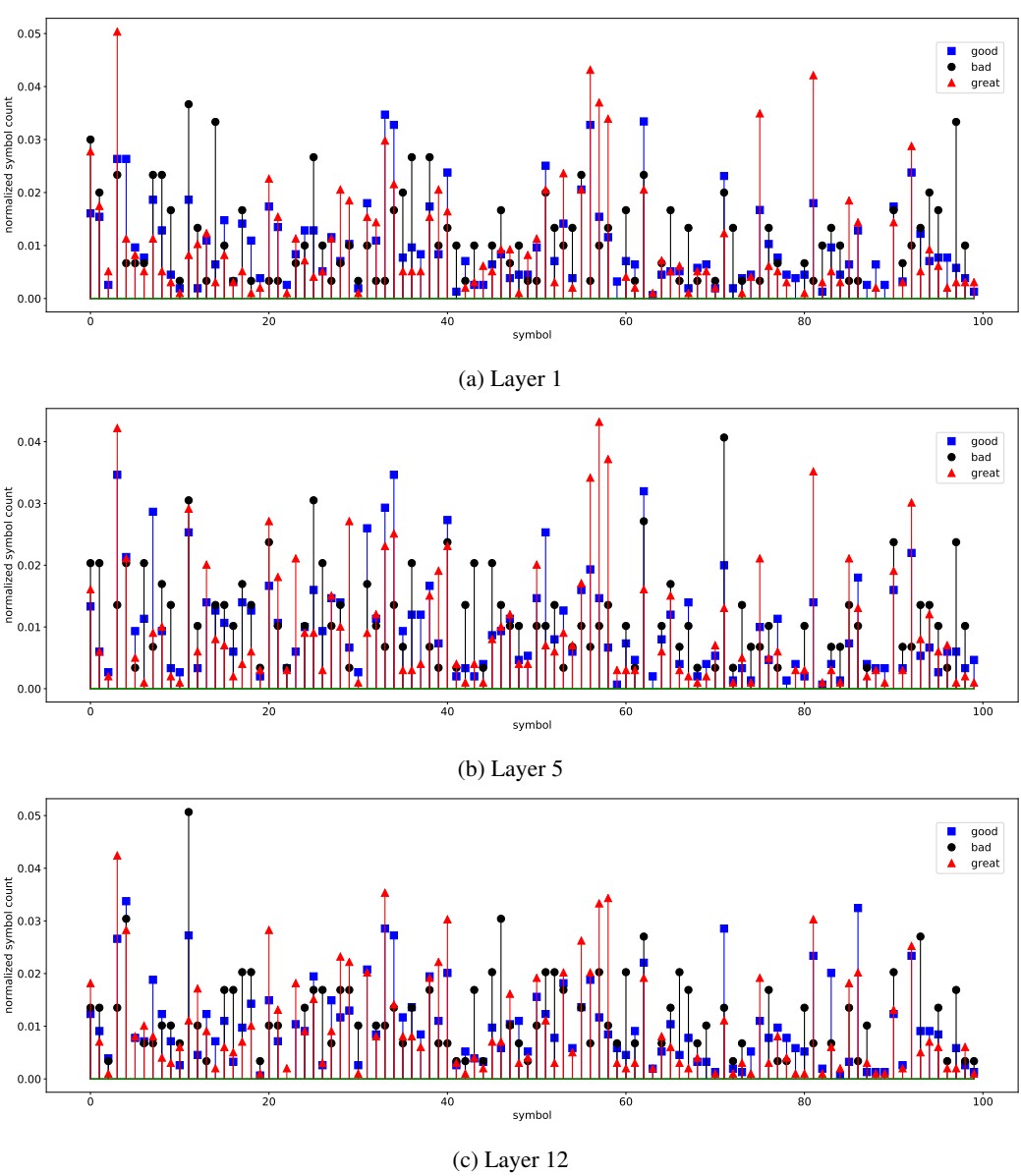

(a) Layer 1

(b) Layer 5

(c) Layer 12

Figure 4: Distribution of most attended symbols when generating tokens *good*, *bad*, *great* for HSN(100, 5) trained on the Yelp data set. The decoder attention matrices between symbols and output are averaged over all attention heads for layers 1, 5 and 12.

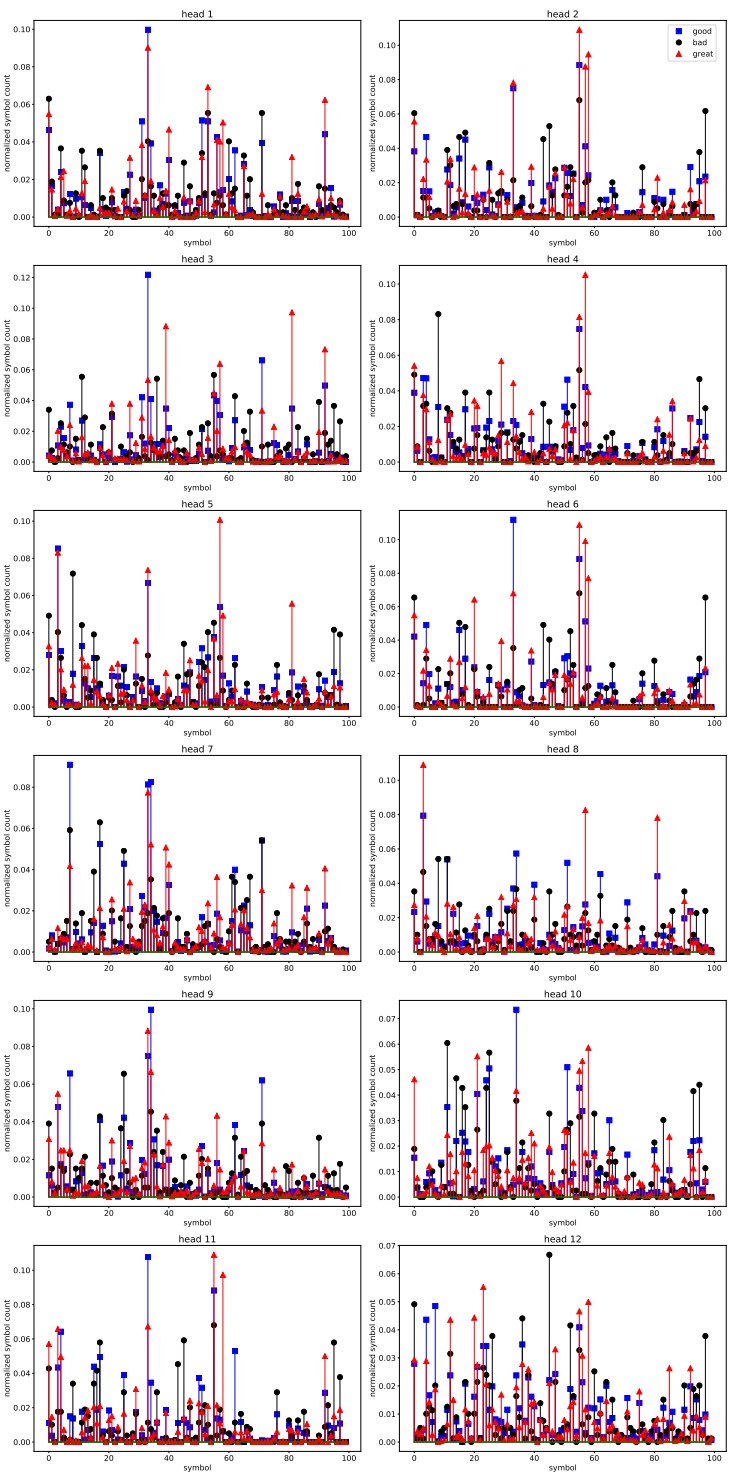

Figure 5: Distribution of most attended symbols when generating tokens *good*, *bad*, *great* for HSN(100, 5) trained on the Yelp data set. The distribution is computed from the decoder attention matrices between symbols and output for each attention head for layer 1.

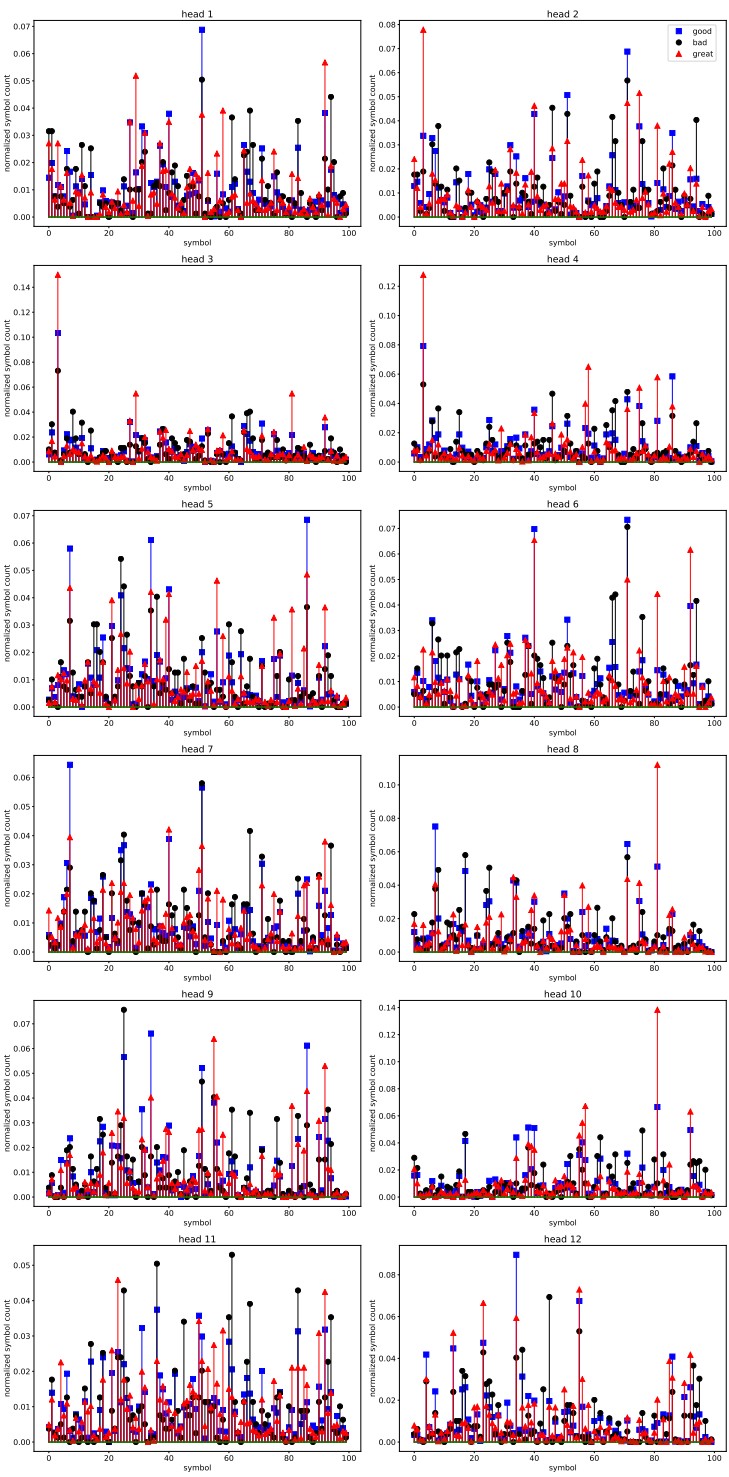

Figure 6: Distribution of most attended symbols when generating tokens *good*, *bad*, *great* for HSN(100, 5) trained on the Yelp data set. The distribution is computed from the decoder attention matrices between symbols and output for each attention head for layer 5.

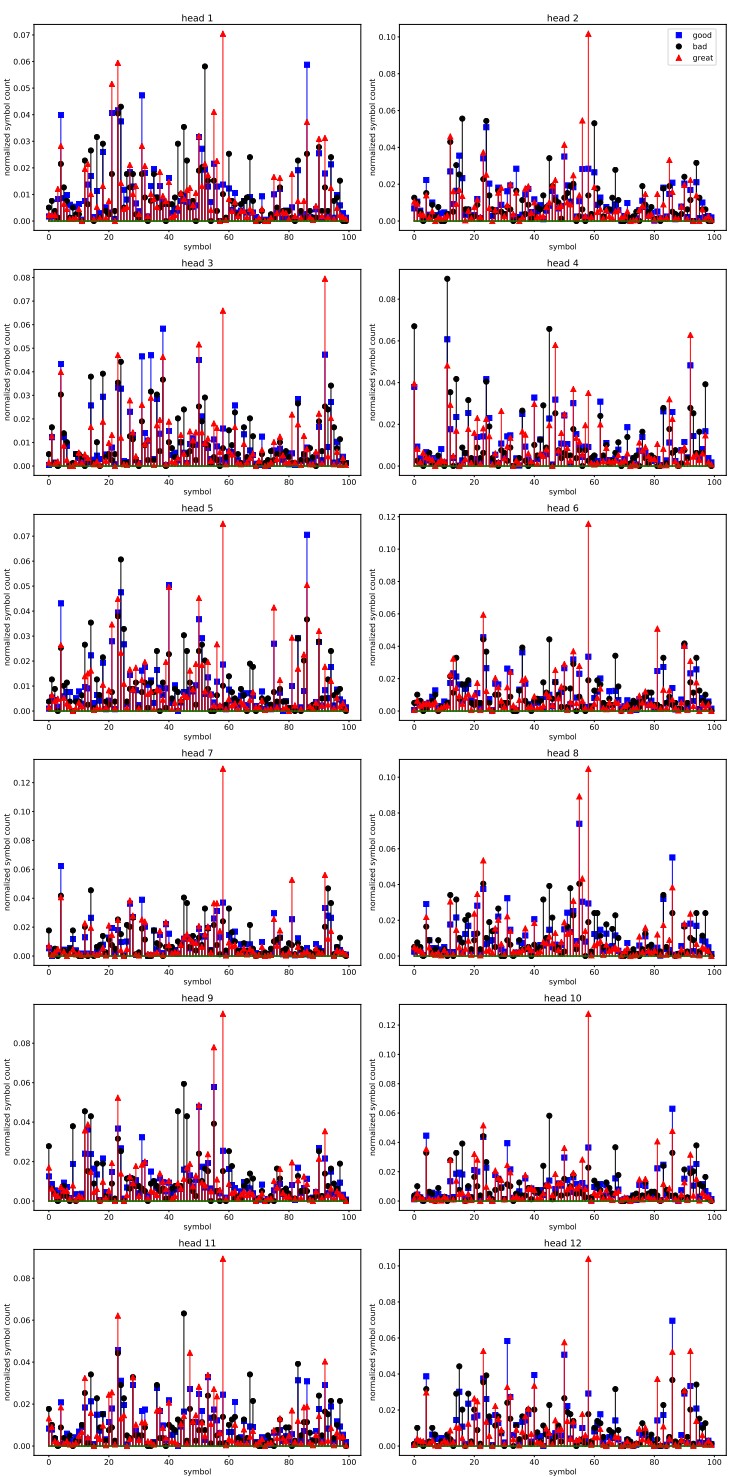

Figure 7: Distribution of most attended symbols when generating tokens *good*, *bad*, *great* for HSN(100, 5) trained on the Yelp data set. The distribution is computed from the decoder attention matrices between symbols and output for each attention head for layer 12.

# G   ON COMMONSENSE REASONING GENERATION

In this section we expatiate on the details of our approach to commonsense reasoning generation.

First, we modify the encoder component of HSN to process the tuples $(\mathbf{s}, \mathbf{r}, \mathbf{o})$ as

$$q_\phi(\mathbf{z}_{1:L}|[\mathbf{s}, \mathbf{r}, \mathbf{o}], \mathbf{A}) = q_\phi(\mathbf{z}_{1:\frac{L}{2}}|[\mathbf{s}, \mathbf{r}], \mathbf{A})q_\phi(\mathbf{z}_{\frac{L}{2}+1:L}|[\mathbf{s}, \mathbf{r}, \mathbf{o}], \mathbf{z}_{\frac{L}{2}}, \mathbf{A}), \tag{39}$$

so that the first half of the schema depends on subject and relation only, whereas the second half depends on the entire 3-tuple. As it will become evident below, this decoupling is necessary for the inference of novel objects.

Each of the posterior distributions above is modelled with the same architecture, as shown in Fig. 1, but sharing a single pretrained BERT model. That is, we have two copies of all pink-shaded blocks in the Fig. 1, one for $q_\phi(\mathbf{z}_{1:\frac{L}{2}}|[\mathbf{s}, \mathbf{r}], \mathbf{A})$, the other for $q_\phi(\mathbf{z}_{\frac{L}{2}+1:L}|[\mathbf{s}, \mathbf{r}, \mathbf{o}], \mathbf{z}_{\frac{L}{2}}, \mathbf{A})$, and a single pretrained BERT model.

Using such a 2-component encoder model we are able to successfully infer schema representations for the KG tuples, as shown in Table 4. The task is however to infer *new objects*, given only subject-relation pairs. We thus need a way to infer schema representations without relying on the phrase object $\mathbf{o}$.

The classical solution to this inference problem is to replace, à la Kalman Filter, $q_\phi(\mathbf{z}_{\frac{L}{2}+1:L}|[\mathbf{s}, \mathbf{r}, \mathbf{o}], \mathbf{z}_{\frac{L}{2}}, \mathbf{A})$ with a local, trainable prior model of the form $p_\theta(\mathbf{z}_{\frac{L}{2}+1:L}|[\mathbf{s}, \mathbf{r}], \mathbf{z}_{\frac{L}{2}}, \mathbf{A})$ – where $\mathbf{z}_{\frac{L}{2}}$ is sampled from $q_\phi(\mathbf{z}_{1:\frac{L}{2}}|[\mathbf{s}, \mathbf{r}], \mathbf{A})$ – and train the prior via the KL term in Eq. 12.

As shown in Section B, maximizing the mutual information between data and representations averages out all local information in the KL term, and thus hinders the learning of the prior – see e.g. HSN[prior] in Table 4: the samples from the prior are not close enough to those of the posterior, hence the significant drop in performance of the model.

An alternative is to train, in the spirit of knowledge distillation (Hinton et al., 2015), a third-party model on the inferred schemata, to predict $\mathbf{z}_{\frac{L}{2}+1:L}$ conditioned on $\mathbf{z}_{1:\frac{L}{2}}$.

Indeed, given the inferred schemata from the training KG, we consider a sequence-to-sequence model which inputs $\mathbf{z}_{1:\frac{L}{2}}$, together with the subject-relation pair, and outputs $\mathbf{z}_{\frac{L}{2}+1:L}$. That is, a model of the form $p_\theta(\mathbf{z}_{\frac{L}{2}+1:L}|\mathbf{z}_{1:\frac{L}{2}}, [\mathbf{s}, \mathbf{r}])$. Specifically we use (i) a bidirectional LSTM network with hidden dimension of 512 to encode the first half of the schemata, (ii) a pretrained BERT model to encode the subject-relation pair, and (iii) a LSTM network of dimension 512 as an autoregressive decoder model. The initial (hidden) states of the latter are determined by an MLP which inputs the representations from the LSTM and BERT encoder models. The model is trained on samples from $q_\phi(\mathbf{z}_{\frac{L}{2}+1:L}|[\mathbf{s}, \mathbf{r}, \mathbf{o}], \mathbf{z}_{\frac{L}{2}}, \mathbf{A})$.

Our preliminary results, HSN[KD] in Table 4, show that this approach improves upon the untrained prior model, and even outperforms the stand-alone COMET(GPT-2) model.

## G.1   ATOMIC DATASET

For this preliminary study we focus only on the ATOMIC dataset of Sap et al. (2019b). It contains 877K $(\mathbf{s}, \mathbf{r}, \mathbf{o})$ tuples covering a variety of social commonsense knowledge around specific *If-Then* events. A bit more in detail, ATOMIC splits its commonsense knowledge into nine categories, covering the event's causes, its effects on the agent, and its effect on other direct (or implied) participants. We use the training splits from Sap et al. (2019b), resulting in 710K training, 80K validation, and 87K test tuples respectively.

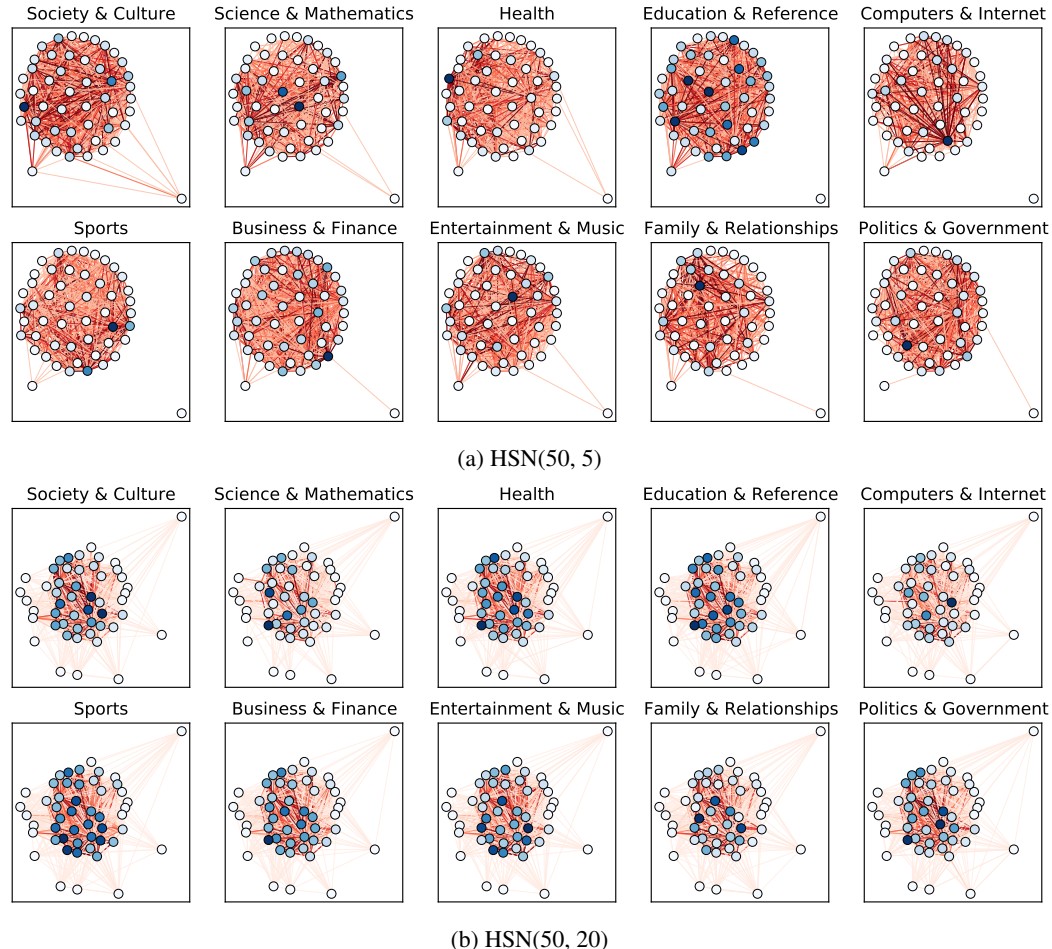

(a) HSN(50, 5)

(b) HSN(50, 20)

Figure 8: Schema distributions inferred from each category of the Yahoo dataset, for HSN(50, $L$) with $L = \{5, 20\}$. The node positions in the figure are consistent among labels and were computed using a force-directed embedding of the global graph $\mathcal{G}$.

---

**with considerable irony the case also shows how completely japan has turned the tables on u.s. business**

(1) in brief the chancellor of the exchequer nigel lawson's decisions were justified by their intended political and financial convenience and credit

(2) analysts said they expect the federal authority to be totally revamped giving japanese manufacturers more clear way to measure their exports.

(3) but others say inco commission has been inadequate

(4) in 1970 banco exterior an agency run by banco exterior <unk> de <unk> <unk> was attempting to reduce liabilities and raise the sale of certain works by the division

**the amended filings also point out that under a new agreement <unk> has an <unk> obligation to sell farmers to axa upon an acquisition of b.a.t**

---

Table 10: Interpolation between two random instances from the PTB dataset

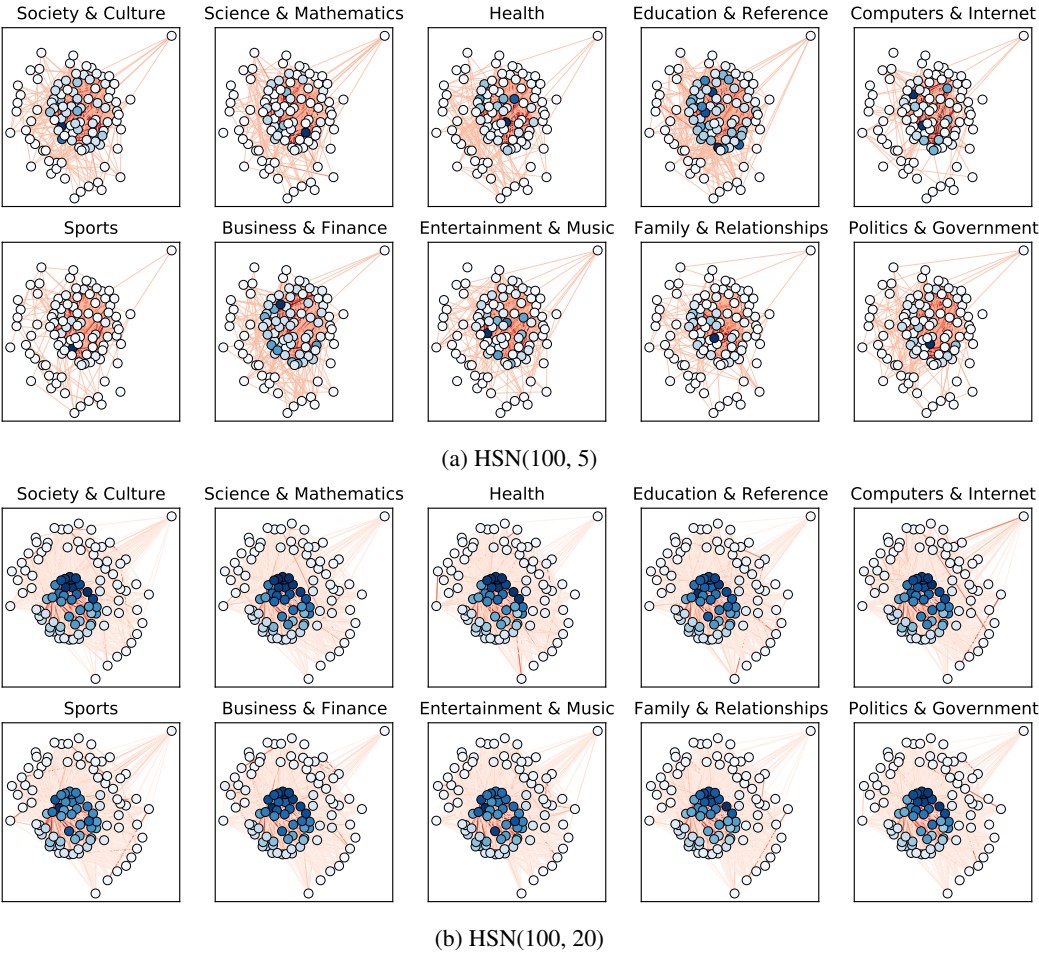

Figure 9: Schema distributions inferred from each category of the Yahoo dataset, for HSN(100, $L$) with $L = \{5, 20\}$. The node positions in the figure are consistent among labels and were computed using a force-directed embedding of the global graph $\mathcal{G}$.

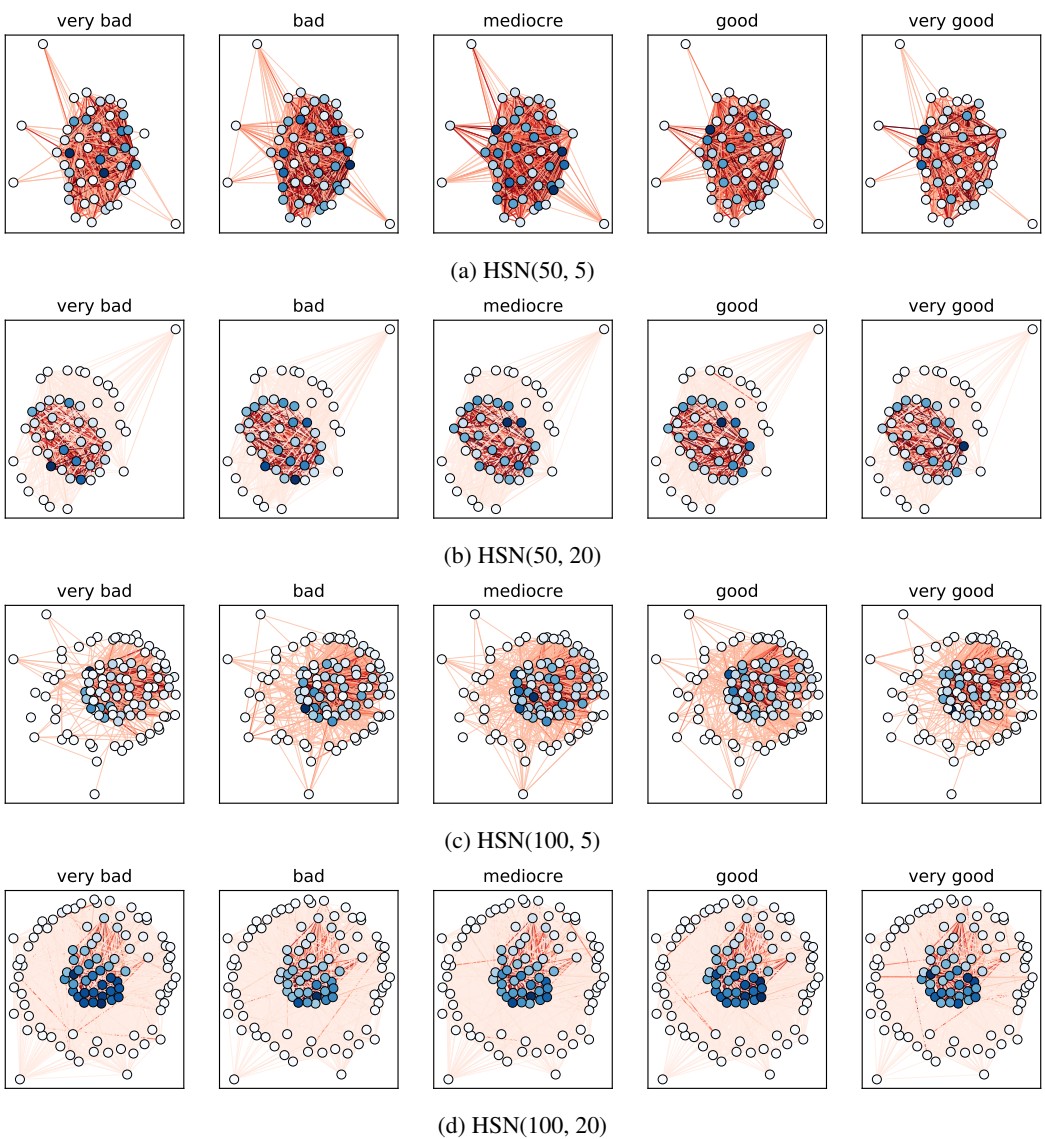

Figure 10: Schema distributions inferred from each category of the Yelp dataset. The node positions in the figure are consistent among labels and were computed using a force-directed embedding of the global graph $\mathcal{G}$.

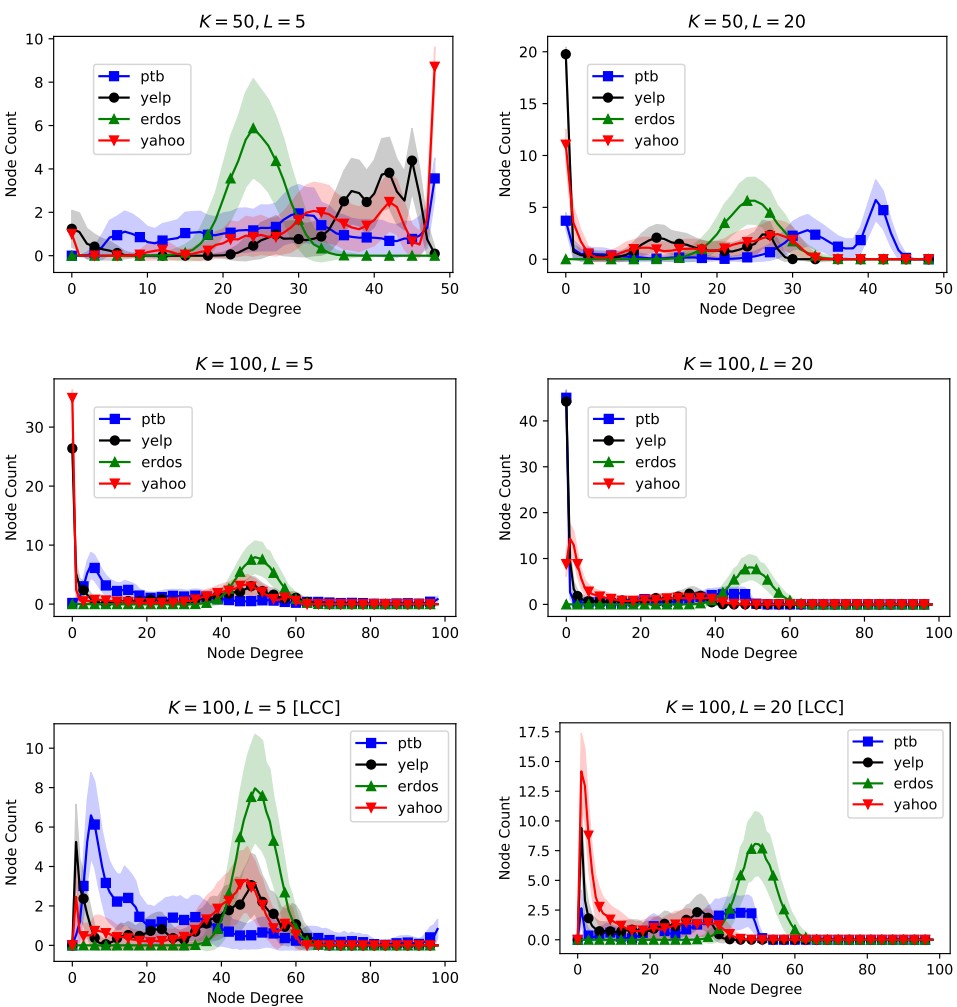

Figure 11: Degree distributions of inferred graphs from all corpora, compared to Erdos-Renyi graphs for $p = 0.5$. The upper four plots show results for full inferred graphs, the lowest two show the degree distributions of the largest connected component of the models for $K = 100$.

| Interpolate Society & Culture Science & Mathematics |
| --- |
| **is steady eye contact good? when i am communicating with someone, i tend to give very steady, long _UNK eye contact. so i tried to _UNK it as a young naive girl, and now it's a habit i can't lose... it just depends on the person you are having the conversation with...** |
| (1) i am 14 years old. their is this girl again who speaks very much of me and talks 2 me, the idea of me 2 her and never gives any suggestion to verbal _UNK for my 2nd. listen, therapy! |
| (2) what do you do when you think your best friend told you shes bisexual? when she says that, or you might have believed if your friend said it is. they're inevitably sharing that they don't share... |
| (3) how do you change liquid in an ice cube into liquid form? paste, mix and freeze _UNK a gallon of co2 into a _UNK and then _UNK in some ice to form a coating. |
| (4) what kind of rules does gravity apply? if a certain weight is placed in a container, the net force applied on it hits the water surface and the right weight will turn into gravity |
| **how does a photovoltaic system that feeds back into the power grid get on the same phase angle? or? should i say does it need to be the same as the _UNK's?** |

| Interpolate: Business & Finance Family – Relationships |
| --- |
| **at 35, am i too old to go to college to become a psychiatrist? i'm 37, and i just started my second semester in a 2 year college... you need to be prepare for the financial aspects, but the social ones are no problem...** |
| (1) what would be a good title for a _UNK _UNK? i have _UNK in _UNK and there are no real courses done for it but i do love the job and i've already done my freshman year. i am currently teaching placement at _UNK and need the same as the average undergraduate student... |
| (2) has anyone here applied in the past 4 months or is it better to get a try out y _UNK a slightly better long term career _UNK ... |
| (3) lately im having trouble with my fiancee, how do i bring him back? it obvious at this point that you can't " bring us together ". try playing games. |
| (4) could i still go out with this guy and still be friends and respect him.? i don't want to just fell in love with the guy that i was with. i want 2 be with friend's girl and still be friends... |
| **how do i know if my man, is inlove with me? well... some questions, how old are _UNK? - are you wealthy?, is he wealthy?, how long have you been together...** |

Table 11: Interpolation between four random instances from the Yahoo dataset

Interpolate: Very bad - bad

**do not use this company!! they told me within one hour, then i called again they said the driver have 90 mins. 90 minutes later, they said the driver is in traffic and wait for 15 minutes, i checked google map no accident, all green on all freeway...**

(1) i ordered for pick up as my daughter hadn't been told that or even ordered online. when i spoke to the young lady, who was _UNK, she carried on a conversation with not a manager. it's bad customer service and i wouldn't even bother with this place...

(2) place was clean... when i called to let them know i 'd get something else, the person that answering the phone wouldn't understand me... really? i gave this restaurant a b + for the cleanliness of the food and the friendliness of the staff

(3) i had the quesadilla and the carnitas tacos. i felt every bite of these were so rubbery and the potatoes were off. i feel like the service and the quality of food can do much better.

(4) somewhat disappointed. i did it once and loved it but today, today's water is bitter and salty... and the mint and cherry blossom _UNK'flavors just taste that way.

**the food quality doesn't match the place at all. i think it's ok for a pub but this place is supposed to be a nice place for professional lunches. i had the chicken flatbread and the chicken was more like subway chicken! with so many options around that area i won't pick this place for lunch.**

Interpolate: Very bad – Very Good

**skip it... there are much better options out there! the " hot " food was not hot, and the flavor was only mediocre at most.**

(1) indifferent to locals. the kids size pizzas were a billion times worse than a pizza hut. the quality of food was just awful. i wouldn't recommend this to a significant other for what it is.

(2) this new mexican spot is ok, bordering on childish. i went with friends and ordered a carne asada burro... it wasn't off the hook ; what made this place great were the chips & salsa sucked. yuck! ...

(3) wow. _UNK you give so much frosting!! we were a groupon special for a cupcake for the princess of chocolate, and we were pretty stoked. they were _UNK and creative. they even suggested we try the coconut ... we 'll definitely be back soon.

(4) went for the first time during a recent trip to vegas. our server jeff made special recommendations for our friends and i. it was fantastic most of the food was light and fresh... i would highly recommend this place!

**i had dinner at republic kitchen tonight for the first time and was very impressed with the service, the decor, the menu, and the food quality... i am going back sunday for their brunch and jazz!**

Table 12: Interpolation between four random instances from the Yelp dataset

