# OpenReview forum: "Hidden Schema Networks"
_ICLR.cc/2023/Conference — Submitted to ICLR 2023_

### Official Review · Reviewer_Dats · 2022-10-24

**Confidence:** 5
**Correctness:** 2
**Technical Novelty And Significance:** 3
**Empirical Novelty And Significance:** 3
**Recommendation:** 3

**Clarity, Quality, Novelty And Reproducibility:**

## Clarity

The formulation of the algorithm in training and generation is still not very clear. It would be better if there are some pseudo-codes to explain the method step by step.

## Novelty

The proposed method is novel.

## Reproducibility

The proposed method has good reproducibility because all resources required for reproduction is already provided in the supplementary material.

**Strength And Weaknesses:**

## Strengths

1. This work studies VAE for natural language. This is an important open problem in NLP and VAE community.

2. They conduct experiments using modern pretrained language models such as BERT and GPT-2, and compared with these strong baselines in experiments.

3. This work presents an interesting analysis on the structure of the latent graph (Figure 3 and Table 3), showing possibility of interpreting latent representations of big language models.

## Weaknesses

1. There is a major soundness problem in the LM evaluation. According to Section 3 and Figure 1, the proposed method first encodes the *full* input sentence using a modified Transformer encoder and get $\mathbf{h}^\text{enc}$, which is used to compute $\mathbf{q}$ and then the node sequence $\mathbf{e}$. In the decoding stage, the modified Transformer decoder model has access to $\mathbf{e}$ via attention. As a result, although indirectly, the **decoder has access to the full input sentence**. An ordinary Transformer encoder-decoder model having access to the full input sentence can easily achieve a perplexity as low as 1 if trained properly. Therefore, the empirical result that the proposed method outperforms GPT in language modeling makes little sense in showing the strength of the proposed method.

2. The formulation of the algorithm in training and generation is still not very clear. It would be better if there are some pseudo-codes to explain the method step by step.

**Summary Of The Paper:**

This paper proposes a variational autoencoder (VAE) framework for encoding natural language a natural language sentence into a sequence of nodes in a global latent graph. The author claims two major contributions:
1. The model enhances the interpretability of traditional pretrained language models by interpreting nodes in the latent graph.
2. The model reaches SOTA performance in language modeling.

**Summary Of The Review:**

The work researches VAE in NLP. It is an open problem of major significance. It proposes HSN to auto-encode text sentences into a trajectory in a latent graph. However, the author claims that HSN outperforms traditional LMs in language modeling task, but I consider this claim questionable because concerns about the soundness of evaluation (see weakness 1 for details). Therefore, I propose to reject this paper unless my concerns are addressed.

---

> ### Author Response · Authors · 2022-11-15
> **Response to Reviewer Dats**
>
> We thank the reviewer for taking the time to review our work and for the helpful comments and criticisms. Below we address all concerns.
>
> **Concerns regarding our evaluation scheme**
>
> Before directly addressing the issue of our evaluation scheme, let us, for the sake of completeness, expatiate somewhat on variational autoencoders for language modeling (VAE-LM). Since first introduced by Bowman et al. (2016), VAE-LM deal with the problem of inferring continuous, sentence-level representations from sequences of word embeddings. In VAE-LM a stochastic encoding function maps the complete sequence of word embeddings composing the sentence in question into a latent variable in some continuous space. A stochastic decoder function then takes the latent representation as input, and processes the same sequence of word embeddings autoregressively, thereby autoencoding it.
>
> Thus, in VAE-LM, the decoder has, at least in principle, always access to information about the full input sentence, **provided the latent variable does in fact encode this information**.
>
> In practice, however, it is nontrivial to autoencode a sentence. Indeed, it is well known that expressive autoregressive decoders tend to ignore the latent code altogether, and rely only on the observed sequence of word embeddings to predict the next word in the sentence. This problem is known as the *posterior collapse problem*, and it is generally understood to be caused by the variational lower bound used to train the autoencoder.
>
> Thus, when researchers evaluate VAE-LM against autoregressive models with no latent variables, they are indirectly validating that the sentence representation does indeed encode useful information (for the reconstruction task) about the sentence. If a VAE-LM outperforms an autoregressive model, it is because the latent representation is successfully being interpreted. That is, because it helps the decoder predict the next word in the sentence.
>
> Please note that testing VAE-LM against language models with no latent variables has been done repeatedly in the NLP literature, and it is by now standard. Some examples follow:
> * VAE models that deploy one-dimensional CNNs decoders (Yang et al. 2017),
> * autoencoders that minimize the Wasserstein distance between original and reconstructed sentences (Zhao et al. 2018),
> * VAEs that map sequences of word embeddings to sequences of discrete symbols (T. Zhao et al. 2018), like we do,
> * VAE models that use BERT and GPT2 as encoder and decoder, respectively (Li. et al. 2020).
>
> In sharp contrast, and in complete agreement with the reviewer, an ordinary Transformer encoder-decoder model (i.e. a translation model) that encodes and decodes the same sentence only needs to implement an identity function. That is, there is a direct signal from the input sequence of the encoder to the decoder model. In VAE-LM, the latent code (be continuous, discrete or a sequence of discrete symbols) **does not** have information about the input sentence from the very beginning. This information has to be encoded.
>
> **Clarity concerns**
>
> We have added a pseudo-code of our algorithm in Appendix C. We hope the latter clarifies further our methodology to infer relational structures that allow for compositionality from sequential data.
>
> **Links to references**
>
> 1. Bowman et al.(2016): https://aclanthology.org/K16-1002.pdf
> 2. Yang et al. (2017): http://proceedings.mlr.press/v70/yang17d/yang17d.pdf
> 3. Zhao et al. (2018): https://arxiv.org/pdf/1706.04223.pdf
> 4. T. Zhao et al. (2018): https://arxiv.org/pdf/1804.08069.pdf
> 5. Li et al. (2020): https://aclanthology.org/2020.emnlp-main.378.pdf

---

> > ### Comment · Reviewer_Dats · 2022-11-19
> > **Thank you for your response**
> >
> > Thank you for your response addressing my concerns. Your newly added Appendix resolved previous concerns about clarity. However, **the evaluation scheme and the contribution of the paper are still in question**.
> >
> > I read all the related works the author listed in the author's response. From these related works, I find that VAE language modeling (VAE-LM) in these works are just used as a *sanity check* to ensure that the VAE indeed learned something. Specifically, by showing that the VAE-LM outperforms simple LM in perplexity, they prove that VAE encodes some useful representations of the sentence. However, it is just a sanity check, a positive result in VAE-LM evaluation *neither* implies the usefulness of such VAE-LM *nor* counts as the main contribution of such a paper. However, in this HSN paper, the author considers a positive result in VAE-LM evaluation as the main contribution of the paper. **The claim in the abstract/intro that "HSN is good because it outperforms GPT-2" is really misleading.**
> >
> > If we exclude the VAE-LM evaluation part from the paper, the only remaining evaluation benchmark is on the ATOMIC dataset (Table 4). In this experiment, HSN is only compared with a single baseline, COMET, and the gain on BLUE-2 is only 0.002. So **the experiment of this paper is much less solid than the related works listed in the response** . For example, OPTIMUS (Li et al., 2020) is evaluated on guided language generation (DailyDialog, Yelp) and low-resource language understanding (the full GLUE benchmark consisting of 8 subtasks) and showed good results. Yang et al. (2017) also evaluate their method on semi-supervised VAE (Yahoo, Yelp) and conditional language generation.
> >
> > In the author's response, the author also claims that VAE is useful because it compresses a sentence or a document into a single latent representation. However, as we talked about before, this claim is supported by insufficient experiments. To my best knowledge, the quality of vector representations of texts is usually evaluated by text tanking or passage retrieval (https://paperswithcode.com/task/passage-retrieval).
> >
> > To summarize, the author does not address my concerns about the evaluation scheme in the paper. **After reading all related works provided in the response, I am even more convinced that the main contribution of the paper is not well supported by the experiments provided in this paper** . Therefore, I do not recommend this paper be accepted because it does not meet the high standards of ICLR.

---

> > > ### Author Response · Authors · 2022-11-22
> > > **2nd response to Reviewer Dats**
> > >
> > > **Language modeling represents neither our end goal nor our main result**
> > >
> > > Roughly following the program outlined by Tenenbaum et al. (2011), and as we tried to make clearer with the revised introduction, our central goal is to impose, via inductive biases, explicit structures and patterns onto the representations learned by large, pretrained language models (LPLM). To be specific, we draw ideas from conceptual role theory (Block, 1986) and conceptual structure theory (Jackendoff, 1978), and seek to impose relational structures which allow for compositionality onto the output representations of LPLM.
> > >
> > > To achieve this goal, we introduced a hidden, global random graph model and assumed that (i) its nodes encode some abstract semantic units of natural language, and that (ii) its edges encode how these semantic units relate to each other. Next, we assumed that natural language sentences can be encoded into short sequences of connected nodes (related semantic units). These sequences are taken to be composed by biased random walkers running along the graph.
> > >
> > > In terms of such modelling assumptions, the task of imposing compositional and relational structures onto the LPLM representations, translates into inferring the posterior distributions over both, the global random graph model, and the local random walk model, conditioned on the LPLM representations.
> > >
> > > Technically, such an inferential problem required an algorithm that mapped the input sequence of tokens into stochastic processes (the random walks), which were in turn conditioned on a global random object (the hidden graph).  We borrowed concepts from the physics literature to parameterize inhomogeneous random walk processes running along graphs with neural networks, and devised a novel, variational inference algorithm to find the posterior distributions that best described the data. To the best of our knowledge, ours is the first algorithm to achieve this task.
> > >
> > > Allows us now to list our contributions in detail:
> > >
> > > 1. We introduced a novel, attention-based neural variational algorithm to infer hidden graph structures, and the random walks running along them, from sequential data.
> > > 2. We empirically showed that the model is able to uncover ground-truth graphs from artificially generated datasets of random token sequences. Likewise, we empirically demonstrated that the model successfully encodes natural language data.
> > > 3. We empirically showed that the model is also able to encode commonsense *In-Then* reasoning data (from left to right, column 4 in Table 4: $HSN$), thereby introducing a novel notion of reasoning as random walk processes running along a semantic graph. To infer novel phrased objects we split commonsense reasoning into two steps: (i) infer the abstract commonsense KG and (ii) generate the second half of the random walk, which encodes plausible phrased objects, from the first half, which encodes the subjects and relations. We succeed in achieving the 1st step (column 4). Preliminary results for the 2nd step are promising (column 6: $HSN_{KD}$).
> > > 4. Motivated by the homotopies between sentence extracted from traditional VAE models, we introduced a notion of *random walk interpolation* and observed that the generated sentences do interpolate between e.g. topics and sentiments (Appendix E.2). Motivated by studies of LPLM, we have also observed that one can infer the meaning that HSN encodes into the schema network, by exploring the attention weights of the decoder (a modified GPT2) model with respect to the symbol sequences. That is, by recording what symbols does the decoder attent to when processing specific words (Appendix F).
> > >
> > > Finally, let us quote Griffiths et al. (2007): ``Learning large-scale representations of abstract conceptual relations from naturally occurring data remains an unsolved problem." (page 3). We believe our contribution is a first step towards the solution of this problem.
> > >
> > > **Links to references**
> > > 1. Tenenbaum et al. (2011): https://cocosci.princeton.edu/tom/papers/LabPublications/GrowMind.pdf
> > > 2. Block, (1986): https://aardvark.ucsd.edu/mind/block_advertisement.pdf
> > > 3. Jackendoff, (1978): https://aclanthology.org/T78-1022.pdf
> > > 4. Griffiths et al. (2007): https://cocosci.princeton.edu/tom/papers/topicsreview.pdf

---

> > > > ### Author Response · Authors · 2022-11-22
> > > > **Response to specific comments in 1st reply of Reviewer Dats**
> > > >
> > > > Two clarifications:
> > > > 1. **We never claimed that** "*HSN is good because it outperforms GPT-2*" **, as written by the reviewer.**
> > > >
> > > > Instead we wrote (in the abstract): "*Quantitatively, our results show that the model successfully interprets the inferred symbol sequences, as it achieves state-of-the-art scores on language modeling benchmarks.*"
> > > >
> > > > This claim was intended to simply mean that HSN successfully encodes the data.  As we wrote above in our 1st response to the reviewer:
> > > >
> > > > "*If a VAE-LM outperforms an autoregressive model, it is because the latent representation is successfully being interpreted. That is, because it helps the decoder predict the next word in the sentence.*"
> > > >
> > > > We nevertheless understand why the claim in the abstract might give the impression that our main goal was language modelling. We hope our latest clarifications in **2nd response to Reviewer Dats** above, together with the revised introduction, clarify what our main goal is.
> > > >
> > > > We shall modify the abstract and remove (or reformulate) the aforementioned claimes.
> > > >
> > > > 2. **We never claimed in our 1st response that** "*VAE is useful because it compresses a sentence or a document into a single latent representation.*"
> > > >
> > > > Our entire comment on the evaluation of VAE-LM was to clarify why researchers compare VAE-LM against LM with no latent variables. As then explicitly written by the reviewer:
> > > >
> > > > "*VAE language modeling (VAE-LM) are just used as a sanity check to ensure that the VAE indeed learned something.*"
> > > >
> > > > This is precisely what we wanted to convey with our reply! **And this is precisely why we compared HSN against GPT2**. Our results in Table 2 demonstrate that HSN indeed encodes something. The subsections titled *Structure of hidden schema networks* and *Schemata and semantics*, as well as the experiments on random walk interpolations (Appendix E), and those on the attention on symbols (Appendix F), aim to explored what precisely is being encoded by the model.

---

### Official Review · Reviewer_nMiy · 2022-10-24

**Confidence:** 4
**Correctness:** 2
**Technical Novelty And Significance:** 2
**Empirical Novelty And Significance:** 2
**Recommendation:** 3

**Clarity, Quality, Novelty And Reproducibility:**

The critics on the language models should be substantially improved to reflect a better understanding and acknowledge of the field, see above discussions.

**Strength And Weaknesses:**

**Strength**

**Interesting generative model.** The generative model proposed in this paper, which discovers a latent schema network within the contextualized representations, is interesting and worth attention.

**Experiments validate the effectiveness of the method.** The experiments is performed in a principled way, starting from a synthetic setting then the real language. The language modeling experiments (though the field have moved from language modeling) shows the advantages of the method

**Weakness**

**Biased critics on language models.** Unfortunately, critics made in this paper towards language models is very biased, or even wrong sometimes. Specifically:

- “lack both compositionality” (abstract and introduction): recent developments in language models, especially when language models are large, show extraordinary ability in generating novel and meaningful sentences compositionally (for example: Brown et. al. 2020, Wei et. al. 2022). The general belief of the community is that the content generated by large language models are quite novel and impressive. The authors can also validate this themselves simply by trying to let GPT3 to generate and play with it (registration is free).
- “lack … semantic interpretability” (abstract and introduction): there is the whole “Bertology” field studying the interpretability, either lexically, syntactically, or semantically (Rogers et. al. 2020, Liu et. al. 2019, Hewitt and Manning 2019). The general belief of the community is that the representations produced by the language models contain very rich semantic interpretations.
- “semantic content is intrinsically relational” (first paragraph in introduction): this statement is wrong. The semantics of certain words may not need to be relational, for example:
    - Proper nouns, like name of countries (the United States). They have semantic meaning on their own, and one does not necessarily need their relations to other words to better understand their meaning.
    - New words that emerges as the society develops. For example, when “electricity” was first discovered. The understanding of these words require grounding to the physical, actual world, but may be hard to understand from their relations to other words.
- “difficult to manipulate” (abstract and introduction): there are multiple works controlling the generation from language models, for example Miao et. al. 2019, Qin et. al. 2022.

**Limited impact of the result:** the field is now moved from language modeling, as it does not necessarily lead to improvements on the actual tasks. So emphasizing on language modeling results may not be a significant enough contribution.

References:

Brown et. al. 2020. Language Models are Few-Shot Learners

Wei et. al. 2022. Chain of Thought Prompting Elicits Reasoning in Large Language Models

Rogers et. al. 2020. A Primer in BERTology: What we know about how BERT works

Liu et. al. 2019. Linguistic Knowledge and Transferability of Contextual Representations

Hewitt and Manning 2019. A Structural Probe for Finding Syntax in Word Representations

Miao et. al. 2019. CGMH: Constrained Sentence Generation by Metropolis-Hastings Sampling

Qin et. al. 2022. COLD Decoding: Energy-based Constrained Text Generation with Langevin Dynamics

**Summary Of The Paper:**

This paper proposes a graphical model parameterized by BERT and GPT2 to infer the hidden schema network within the language data. Experiments show that this model is effective in recovering the underlying graphs (to an extend) and effective in language modeling.

**Summary Of The Review:**

At the first sight of this paper, in the introduction and related work, I was surprised by the critics in this paper made towards language models (”lack both compositionally and semantic interpretability”; “difficult to manipulate”) which is simply wrong (at least from my understanding of the field and years as an NLP researcher). Yet when I continue to read the modeling part, the inference method, and the experiment setting, I find these content seem to be OK (from a modeling perspective). I would refrain myself from rejecting a paper because lack of commonsense of the area (which in this case is somehow independent of its tech aspects). Yet I would also refrain myself from accepting a paper that has a wrong understanding of its own filed.

Overall, I think the critics in the paper (abstract, introduction, and related work) requires substantial revision to show, understand and acknowledge the development of language models. Yet apart from that, the modeling seems to be OK but requires stronger results, more than language modeling. Combining these aspects, I rate this paper as a reject.

---

> ### Author Response · Authors · 2022-11-15
> **Response to Reviewer nMiy**
>
> We thank the reviewer for the thorough review.
>
> The main weaknesses pointed out by the reviewer are
>
> 1. **Biased criticism on Large, pretrained language models (LPLM)**
>
> We agree with the reviewer that LPLM can generate content that is novel and impressive. Likewise, we agree that these models encode rich semantic and syntactic content. In fact, we completely concur with the statements made in Rogers et al. (2020), cited by the reviewer, which explain that most of the semantic and syntactic properties of LPLM, albeit rich, are inferred only a posteriori. Our main goal is thus not to criticize but rather to build on top of LPLM, and to translate their **implicit** semantic and syntactic content into **explicit** structures that connect back with classical notions of linguistics.
>
> With these thoughts in mind we have:
> * removed the starting sentence of our abstract
> * replaced, in the introduction, the sentence ``Yet, these representations lack semantic interpretability..." with "Yet, it is still unclear whether such representations can be composed into representations of novel sentences. In fact, most of their syntactic properties are
> implicit..."
> * replaced, in section 3, the sentence ``...these models lack both semantic and syntactic interpretability..." with "Yet, most of the linguistic structure encoded by the output representations of these models is implicit and difficult to interpret"
>
> 2. **Limited impact of our results**
>
> We would like to stress that our aim is not that of language modelling in the classical sense, but to explicitly enforce, through inductive biases, certain qualities or characteristics on the representations obtained by LPLM, in order to improve their performance, while simultaneously introducing a notion of reasoning in a fully unsupervised manner.
>
> Although is true that some notions of compositionality and relations between the output representations of LPLM can be identified without the use of our procedure, the identification is done **a posteriori**, and usually with probes and other mechanism which are intended only as a test of the behavior of the models. Quoting one of the references offered by the reviewer: ``the fact that a linguistic pattern is not observed by our probing classifier does not guarantee that it is not there, and the observation of a pattern does not tell us how it is used" (Rogers et. al. 2020). Following precisely this point, we enforce representations that **a priori**
> * guarantee that some patterns are present in the model, independent of any probe
> * define how such patterns are in fact used.
>
> Referring again to Rogers et al. (2020): ``BERT cannot reason based on its world knowledge". This problem is what motivates studies the likes of COMET (Antoine Bosselut et al., 2019) which require the relationship among instances to be specified in the data, i.e. as a form of supervision. We use our methodology to map such relationships onto a graph of representations, which naturally allows us to introduce notions of reasoning (i.e. as random walks processes along the graphs).
>
> **Other concerns**
>
> *On compositionality:* one typically finds two different senses of compositionality in the literature. One of these concerns the models themselves, instead of their output representations, and regards (or defines) models as compositional, if they can learn compositional solutions. The work of Hupkes et al. (2020) focuses on this sense of compositionality. Likewise, the works of Brown et al. (2020) and Wei et al. (2022), cited by the reviewer, provide evidence for functional compositionality. The other sense of compositionality regards the compositionality of representations. This is precisely the kind of compositionality we refer to in our paper. Indeed, given a sentences (which is composed of a sequence of words), we infer sentence representations which are a priori (i.e. by construction) compositional, for they are built from primitive units (our symbols). In contrast, it is unclear whether LPLM output representations can be composed into representations of novel sentence (Yu & Ettinger, 2020; Bhathena et al., 2020).
>
> *On semantic content being intrinsically relational*:  this statement is, of course, not a novel claim we are making. On the contrary, it is part of a long, still on-going debate in philosophy and linguistics. A classical reference is the language-game concept in Wittgenstein (1953). See also Block (1986) and Margolis and Laurence (1999). Piantadosi and Hill (2022) even argue for (*implicit!*) relational semantics in LPLM. Notwithstanding note that, in our introduction, we are careful to include not only concepts, but also words and percepts in the set of knowledge units which interact and compose semantic content. Thus, strictly speaking, our wording does not have problems with the examples given by the reviewer (i.e. proper nouns and words requiring grounding), even if these can be accounted for by relational semantics (see e.g. Block,1986).

---

> > ### Author Response · Authors · 2022-11-18
> > **References (in response to reviewer nMiy)**
> >
> > **Links to references**
> >
> > 1. Hupkes et al. (2020): https://arxiv.org/abs/1908.08351
> > 2. J. Andreas (2019): https://openreview.net/pdf?id=HJz05o0qK7
> > 3. Yu and Ettinger (2020): https://aclanthology.org/2020.emnlp-main.397.pdf
> > 4. Bhathena et al. (2020): https://aclanthology.org/2020.repl4nlp-1.22.pdf
> > 5. Rogers et. al. (2020): https://aclanthology.org/2020.tacl-1.54.pdf
> > 6. Antoine Bosselut et al. (2019): https://aclanthology.org/P19-1470.pdf
> >   7. Wittgenstein (1953): http://mickindex.sakura.ne.jp/wittgenstein/witt_pu_gm.html
> >   8. Block (1986): https://aardvark.ucsd.edu/mind/block_advertisement.pdf
> >   9. Margolis and Laurence (1999): https://mitpress.mit.edu/9780262631938/concepts/
> >  10. Piantadosi and Hill (2022): https://openreview.net/forum?id=nRkJEwmZnM

---

### Official Review · Reviewer_5yft · 2022-10-28

**Confidence:** 3
**Correctness:** 3
**Technical Novelty And Significance:** 3
**Empirical Novelty And Significance:** 3
**Recommendation:** 8

**Clarity, Quality, Novelty And Reproducibility:**

* Reproducibility - Unclear if the perplexity calculation in Table2 is using the same evaluation code for all models considered. It is possible that there are minor differences (like removal of stop words) in the evaluation technique which are causing the errors.
* Clarity -  Sections of the paper could be more clear. It would be better if authors can provide a better qualitative analysis and a description of Figure1 which is aligned with the math in Section3. It would be better if it is made clear which component is producing which probability distribution.
* Quality - The paper is well written with a good flow and very few grammatical issues.
* Novelty - The work is novel in the sense that it develops a discrete symbol based approach for learning a representation for text, which allows for compositionality.

**Strength And Weaknesses:**

**Strengths:**
1. Authors present a generative model which is better than GPT2 on the Language modelling task.
2. Presents a scientifically sound proof of concept which shows that the network indeed learns the latent graph representation, and that the results are not simply due to randomness.
3. Impactful problem addressed: Capturing relational and compositional features of semantic contents means that these representations can prove to be better for use in common NLP tasks which right now rely on contextual encodings (like BERT encodings).

**Weaknesses:**
1. Authors claim that the contextual sentence embeddings lack semantic interpretability. Authors claim is that since different sets of "hot" symbols represent different topics (e.g. Science and Math), HSN provides semantic interpretability. However, can't we say similar things for continuous contextual features? It's highly possible that in a similar way "Science And Math" topic has high value for some specific continuous features, whereas "Health" has high values for some other set of features. In this way, the significance of the work (of using discrete symbols) is not clear.
2. The authors make claims about compositionality, but they do not compare the compositionality using existing metrics which have been studied in the literature. https://aclanthology.org/2020.repl4nlp-1.22.pdf. Since compositionality is a major part of the paper, readers would be interested in how the compositionality compares to BERT.
3. It is not clear why enforcing a semantic understanding for the (hidden representation) symbols significantly helps with the language modelling performance. At the end of the day, the hidden representation is an internal detail. More text is needed on this aspect.

**Summary Of The Paper:**

The paper proposes building a generative LM (HSN) using a VAE framework to encode sentences into symbols which encode different aspects of language (like topics and sentiments). Using these discrete symbols, the authors claim that we can achieve compositionality of sentence encodings.

The proof of concept is demonstrated by using HSN to generate a sequence of symbols which correspond to a known ground truth latent graph (nodes corresponding to symbols). A significant overlap is seen between the ground truth graph and the generated graph, implying that the model is successful in uncovering the latent graph.

The authors then show the performance of HSN while using BERT and GPT2 to train the model for LM and Commonsense Generation task. The performance (perplexity) of HSN is better than other baselines which is impressive. Different topics of text sentences are represented by different "hot" symbols in the latent graph.

**Summary Of The Review:**

The paper proposes a novel approach for representation learning which allows for compositionality. This approach leads to significant improvements in LM performance over other predominant models like GPT. The paper is mostly thorough in its analysis, but some of the claims regarding compositionality should be better justified using existing mathematical metrics comparing different representations. The paper could however explain qualitative aspects in a better manner. Apart from a few minor things, it is well written.

---

> ### Author Response · Authors · 2022-11-15
> **Response to Reviewer 5yft**
>
> We would like to thank the reviewer for both the detailed review and the kind words about our work. Below we address each of the mentioned concerns and weaknesses in detail.
>
> **Concerns about interpretability claims**
>
> The interpretability of representations obtained via language models is typically illustrated by explicitly looking at the relationships between these representations in an ad hoc fashion: one chooses similar words (according to human notion of similarity) and checks whether their representations are also similar (wrt some metric). Naturally, and as pointed out by the reviewer, this works for both continuous and discrete representations, and we have used such heuristics in the subsection titled ``Schemata and semantics" of section 5 of our paper.
>
> The reason for our claim is, however, that arguably it is easier for a human to attach meaning to a vocabulary of symbols, than to a high-dimensional continuous space. Furthermore, just as researchers investigate the distribution of attention weights in Transformer models, we can investigate the distribution of attention weights of the decoder in HSN, over the symbols, as it processes specific words. Such a procedure could help humans infer the meaning HSN encodes into the schema network.
>
> To exemplify this point we have added a new section (Appendix F) to the paper. There we examine to which symbols does the decoder of HSN, trained on Yelp, attend to, when processing the words *good*, *great* and *bad*. Interestingly, the distribution of symbols that are attended to when processing the word *great* is closer to the distributions of symbols attended by the word *good*, than to the distributions of symbols attended by the word *bad*.
>
> To avoid misunderstanding we have, nevertheless, rephrased our remarks about contextual sentence embeddings lacking semantic interpretability. Please see our reply to reviewer nMiy for details.
>
> **Concerns about compositionality claims**
>
> As pointed out already by Hupkes et al. (2020), there are, in the machine learning community, different notions (or definitions even) of compositionality in place. One such notion understands models to be compositional if they can learn *compositional solutions*. That is, models are presented with tasks which encode some aspects of compositionality, and are assumed to implement compositional functions if they successfully solve the task in question. Following Pelletier (2011), we refer to this class of compositionality as functional. Note that the main contribution of the work by Bhathena et al. (2020), cited by the reviewer, and that by Wei et al. (2022), cited by reviewer nMiy, either test or provide evidence for *functional* compositionality.
>
> Throughout our work we refer instead to *compositionality of representations*. That is, given a sentence (which is composed of a sequence of words), we infer sentence representations which are a priori (i.e. by construction) compositional, for they are built from primitive units (our symbols).
>
> We have added the following footnote to the manuscript:
>
> ``Note that throughout the paper we refer only to compositionality of representations and not to the compositional functions that can be implemented by the models we use. The latter, functional compositionality, is studied by e.g. Hupkes et al. (2020)."
>
> and hope it will help avoid confusions.
>
> That being said, we also plan to investigate how the explicit compositional character of the schemata affects the functional compositionality of HSN, as compared with that of e.g. GPT2, following the framework of Hupkes et al. (2020).
>
> **Concerns on the effect of our representations on language modelling performance**
>
> Although we are not certain about this in the context of language modelling, one of the fundamental advantages of graph modelling is that of missing edge completion. HSN has the ability to infer missing connections, not seen during training, that might account for unseen symbol combinations, and hence for unseen sentences.
>
> This is the logic behind the atomic dataset completion task, where reasoning is akin to uncovering such unseen connection.
>
> We speculate that performance might have improved because the random walk model can generate novel representations at inference time, albeit within the structure uncovered during training.
>
> **Concerns on reproducibility**
>
> We have added a small sentence to the caption of Table 2 which reads: *End-of-sequence tokens are kept during evaluation*.
>
> Although not explicitly written in the work of  Li et al. (2020), we have checked in their publicly available code ( https://github.com/ChunyuanLI/Optimus) that they too keep the end-of-sequence token during evaluation. We can therefore safely compare against their results.
>
>
> **Links to references**
>
> 1. Hupkes et al. (2020): https://arxiv.org/abs/1908.08351
> 2. Bhathena et al. (2020): https://aclanthology.org/2020.repl4nlp-1.22.pdf
> 3. Li et al. (2020): https://aclanthology.org/2020.emnlp-main.378/

---

### Official Review · Reviewer_tUks · 2022-10-29

**Confidence:** 3
**Correctness:** 3
**Technical Novelty And Significance:** 4
**Empirical Novelty And Significance:** 4
**Recommendation:** 8

**Clarity, Quality, Novelty And Reproducibility:**

The writing is clear and the paper provides all details for reproducibility. The method is novel and the results are surprising.

**Strength And Weaknesses:**

*Strengths*

The proposed method has the potential to be more interpretable and controllable than single-vector representations and achieves strong performance on ELBO evaluations. The perplexity bound evaluations in this paper are standard in the related sentence VAE literature [1,2].
Before reading this paper, my prior belief was that such a latent variable model would not result in perplexity improvements. Those beliefs have now changed.

*Weaknesses*

The paper is convincing but could be improved with further experiments on controllable text generation (following Li et. al 2020) and further development of the commonsense reasoning experiment.

Edit: I also agree with reviewer 5yft's position that compositionality claims are not supported. Reducing the claim to a state-of-the-art sentence VAE with discrete representations is one solution.

*Questions and comments*
* To check my understanding, is the following alternative interpretation of the model correct: the model is a (discrete, first-order) hidden Markov model with an autoregressive emission model (GPT-2) and global sparsity distribution over the transition matrix (schema network)?
* What is the reason for using LSTMs in the commonsense reasoning schema completion model (Appendix E)? BART or GPT2 might give better results.

[1] Li, Chunyuan et. al. “Optimus: Organizing Sentences via Pre-trained Modeling of a Latent Space.” EMNLP (2020).

[2] Kim, Yoon et al. “Semi-Amortized Variational Autoencoders.” ICML (2018).

**Summary Of The Paper:**

The paper proposes a method for learning discrete sequential representations of sequence data. The model demonstrates surprisingly strong performance in language modeling, and preliminary results on commonsense reasoning seem promising. Analysis of the schema networks also demonstrates interpretability.

**Summary Of The Review:**

I recommend acceptance. The paper delivers on the surprising claim of a state-of-the-art sentence VAE with discrete latent representations.

Edit on 12/5: After further discussion with the authors, the perplexity comparisons do not use a valid bound. Without updated and fair numbers, I am changing my score to reject until the evaluation numbers are finalized. I think this is interesting work and encourage the authors to (re-)submit with updated numbers and claims.

Edit on 12/13: After receiving the updated evaluation metrics, I am changing my score back to accept.

---

> ### Author Response · Authors · 2022-11-15
> **Response to Reviewer tUks**
>
> First of all, we would like to thank the reviewer for the helpful comments and questions. We very much appreciate your recognition of the significance of our contribution. Below we address your questions:
>
> **Can one interpret HSN as a hidden Markov model?**
>
> yes, HSN can indeed be interpreted as a hidden Markov model (hMM) with an autoregressive emission model, and with the global random graph model masking out (some of) the entries of the transition matrices, provided it is understood that:
> * different from classical hMM, the autoregressive emission model has access at all times to the entire sequence of symbols (schema), and in particular that the length of the latter can differ from the length of the sentence being decoded;
> * different from classical hMM, and similar to VAE, the transition matrices are local (i.e. there is a sequence of transition matrices **per each** sentence) during inference (encoder model), and can be either local or global during prediction (prior random walk model). Indeed, note that the prior model for the commonsense reasoning task generates random walks conditioned on the data point of interest, and thus is local. In contrast, we use a global prior random walk model for the language modelling task.
>
> **Why do we use LSTMs in the knowledge distillation from the schema representations for common sense reasoning?**
>
> We chose an LSTM for simplicity. In practice any sequence-to-sequence model can be used in its place, and we agree with the reviewer that using Transformer-based models might improve the quality of the distilled knowledge. We plan to explore these directions in our follow up contributions.
>
> **Concerns of compositionality**
>
> We kindly ask the reviewer to refer to our reply to reviewer 5yft regarding our claims on compositionality.

---

> > ### Comment · Reviewer_tUks · 2022-11-20
> > **Evaluation, inference, and loss details**
> >
> > Thanks for the response. I have a couple questions about evaluation and perplexity:
> > 1. Is $\mathcal{L}$ (Eq 12), the sum of the ELBO + mutual information, guaranteed to be a lower bound on the marginal likelihood?
> > 2. Is the perplexity estimated by approximating equation 12 or 15, and how many samples?

---

> > > ### Author Response · Authors · 2022-11-22
> > > **2nd response to Reviewer tUks**
> > >
> > > **Is Eq. 12 guaranteed to be a lower bound on the marginal likelihood?**
> > >
> > > No. Let us label Eq. 15 with ${\cal L}\_{elbo}$ and Eq. 12 with ${\cal L}\_{HSN}$. As shown in Appendix B.3
> > >
> > > ${\cal L}\_{elbo} + I(\mathbf{x}\_{1:T}, \mathbf{z}\_{1:L}|\mathbf{A}) = {\cal L}\_{HSN}$.
> > >
> > > That ${\cal L}\_{elbo}$ is a lower bound on the marginal log likelihood follows, as usual, from
> > >
> > > $\log p\_{\theta}(\mathbf{x}\_{1:T})  = \log \sum\_{\mathbf{z}\_{1:L}} \sum\_{\mathbf{A}}  q\_{\phi}(\mathbf{z}\_{1:L}, \mathbf{A}|\mathbf{x}\_{1:T}) \left(\frac{p\_{\theta}(\mathbf{x}\_{1:T}, \mathbf{z}\_{1:L}, \mathbf{A})}{q\_{\phi}(\mathbf{z}\_{1:L}, \mathbf{A}|\mathbf{x}\_{1:T})}\right) \ge \sum\_{\mathbf{z}\_{1:L}} \sum\_{\mathbf{A}}  q\_{\phi}(\mathbf{z}\_{1:L}, \mathbf{A}|\mathbf{x}\_{1:T}) \log \left(\frac{p\_{\theta}(\mathbf{x}\_{1:T}, \mathbf{z}\_{1:L}, \mathbf{A})}{q\_{\phi}(\mathbf{z}\_{1:L}, \mathbf{A}|\mathbf{x}\_{1:T})}\right) \equiv {\cal L}\_{elbo},$
> > >
> > > where we used Jensen's inequality with respect to the expectation value of $q\_{\phi}(\mathbf{z}\_{1:L}, \mathbf{A}|\mathbf{x}\_{1:T})$, which is defined in Eq. 7.
> > >
> > > Therefore $\log p\_{\theta}(\mathbf{x}\_{1:T}) + I(\mathbf{x}\_{1:T}, \mathbf{z}\_{1:L}|\mathbf{A}) \ge {\cal L}\_{HSN}$, which means Eq. 12 is not a lower bound on the log-likelihood.
> > >
> > > Note that we can similarly derive another bound on the log-likelihood, if we use instead Jensen's inequality with respect to the expectation value of
> > >
> > > $q^{\*}\_{\phi}(\mathbf{z}\_{1:L}, \mathbf{A}) = q^{\*}\_{\phi}(\mathbf{z}\_{1:L}| \mathbf{A}) q\_{\phi}(\mathbf{A})$,
> > >
> > > with $q^{\*}\_{\phi}(\mathbf{z}\_{1:L}| \mathbf{A})$ the data-aggregated posterior distribution over random walks.
> > >
> > > This second  lower bound on the log-likelihood of the data can be explicitly written as
> > >
> > > $$
> > > {\cal L} = \sum\_{\mathbf{z}\_{1:L}} \sum\_{\mathbf{A}}  q^{\*}\_{\phi}(\mathbf{z}\_{1:L}| \mathbf{A})q\_{\phi}(\mathbf{A}) \log p\_{\theta}(\mathbf{x}\_{1:T} |\mathbf{z}\_{1:L}) - \sum_{\mathbf{A}} q_\{\phi}(\mathbf{A})  KL[q^{\*}\_{\phi}(\mathbf{z}\_{1:L}| \mathbf{A}); p(\mathbf{z}\_{1:L}| \mathbf{A})] - KL[q\_{\phi}(\mathbf{A}); p(\mathbf{A})].
> > > $$
> > >
> > > This last expression is however  *not* equivalent to Eq. 12; they differ on the 1st (reconstruction) term.
> > >
> > > **Is the perplexity estimated by approximating equation 12 or 15?**
> > >
> > > We nevertheless approximate the perplexity with Eq. 12. That is, with $\exp(-{\cal L}\_{HSN}/T)$. Furthermore, we approximate the expectation values in the reconstruction term of Eq. 12 with a single sample from the posterior distributions. These samples correspond to samples from the aggregated posterior (i.e. via ancestral sampling). Therefore we take Eq. 12 thus computed to be an approximation of the second bound ${\cal L}$.
> > >
> > > Note that other VAE language models that also maximize the mutual information choose to report instead the *reconstruction perplexity*: the exponential of the reconstruction term divided by the number of words in the sentence. This reconstruction perplexity is clearly always smaller than $\exp(-{\cal L}\_{HSN}/T)$. Therefore using the *reconstruction perplexity* does not change our conclusions from Table 2, but makes the gap between HSN and the baselines larger.
> > >
> > > For completeness we will add to the Appendix a new table displaying:
> > > 1. the reconstruction error,
> > > 2. the KL terms,
> > > 4. the mutual information between data and representation,
> > > 3. the perplexity computed via ${\cal L}\_{elbo}$ (Eq. 15),
> > > 4. the perplexity computed via the second elbo ${\cal L}$.
> > >
> > > All but the last one are also available in the papers by Li. et al. (2020) and Fang et al. (2019), so explicit comparisons can be made with respect to all these quantities.
> > >
> > > **... and how many samples?**
> > >
> > > In practice, and as we wrote above, the expectation values in the reconstruction term of Eq. 12 are approximated with a single sample from the posterior distributions, and are then averaged over mini-batches of size 32. The KL terms, in contrast, have closed form expressions.

---

> > > > ### Comment · Reviewer_tUks · 2022-11-26
> > > > **Perplexity bound**
> > > >
> > > > Li et. al (2020) [1] report an importance-weighted lower bound (IWAE [2]), in addition to reconstruction PPL and other metrics following the evaluation in He et. al. [3]. Including metrics 1-5 in the appendix for completeness is a great idea. Please also ensure that the perplexity comparison in the main text uses a valid bound when comparing to language models without latent variables.
> > > >
> > > > 1) Using a valid bound, is there still a perplexity improvement over GPT-2? It would be great to include that number either way.
> > > > 2) Additionally, does HSN still improve over other VAEs when comparing perplexity bounds computed in the same way? Comparing reconstruction perplexity is not enough because of the posterior-prior KL term in the ELBO.
> > > >
> > > > If 1) is no longer true but 2) is at least comparable to the baselines, the method is novel and my score would remain unchanged as long as the numbers are reflected in the claims. However, without a valid comparison I cannot recommend acceptance.
> > > >
> > > > A missing related work: Stratos and Wiseman (2020) [4] also learn a fixed-length structured (Markov) discrete latent code, but by optimizing mutual information.
> > > >
> > > > [1] Li, Chunyuan et al. “Optimus: Organizing Sentences via Pre-trained Modeling of a Latent Space.” Conference on Empirical Methods in Natural Language Processing (2020).
> > > >
> > > > [2] Burda, Yuri et al. “Importance Weighted Autoencoders.” ICLR 2016.
> > > >
> > > > [3] He, Junxian et al. “Lagging Inference Networks and Posterior Collapse in Variational Autoencoders.” ICLR 2019.
> > > >
> > > > [4] Stratos, Karl and Sam Wiseman. “Learning Discrete Structured Representations by Adversarially Maximizing Mutual Information.” ICML 2020.

---

> > > > > ### Author Response · Authors · 2022-12-12
> > > > > **On perplexity calculations**
> > > > >
> > > > > We have recomputed the perplexity (ppl) of HSN via the "importance weighted" Monte Carlo estimator of the marginal log-likelihood
> > > > >
> > > > > $ \log p_{\theta}(\boldsymbol x_{1:T}) = \log \frac{1}{R} \sum\_{r}^R \frac{1}{S}  \sum\_{s}^S \frac{p_{\theta}({\boldsymbol x}\_{1:T}| {\boldsymbol z}\_{1:L}^{(r)}) p({\boldsymbol z}\_{1:L}^{(r)}|A^{(s)})p(A^{(s)})}{q\_{\phi}({\boldsymbol z}\_{1:L}^{(r)}|A^{(s)})q\_{\phi}(A^{(s)})}$.
> > > > >
> > > > > The new results, together with the reconstruction cost (Rec), Kullback-Leibler divergence (KL) and mutual information (MI) values, follow:
> > > > >
> > > > > 1. PTB
> > > > > Model  |     ppl  |   Rec       | KL    | KLg       | MI |
> > > > > :------:  | :------: |     :------:  | :------: |  :------: | :------: |
> > > > > GPT-2 | 24.23 |- | - | - | -|
> > > > > iVAE | 53.44 | 74.69 | 12.51 | - | 12.50
> > > > > optimus ($\lambda=0.05$)| **23.58** |  86.43 | 4.88 | - | 3.78
> > > > > optimus ($\lambda=1.0$)| 35.53 |  77.65 | 28.50 | - | 8.18
> > > > > HSN | 31.73 | 76.79 | 27.91 | 0.49 | **26.06**
> > > > >
> > > > > 2. Yahoo
> > > > > Model  |     ppl  |   Rec       | KL   | KLg       | MI |
> > > > > :------:  | :------: |     :------:  | :------: |  :------: | :------: |
> > > > > GPT-2 | **22.00** |- | - | - | -|
> > > > > iVAE | 47.93 | 297.70 | 11.40 | - | 10.70
> > > > > optimus ($\lambda=0.05$)| 22.34 |  282.84 | 6.97 | - | 5.34
> > > > > optimus ($\lambda=1.0$)| 24.92 |  270.80 | 30.41 | - | 9.18
> > > > > HSN |  22.98 | 268.22 | 24.14 | 0.006 | **22.81**
> > > > >
> > > > > 2. Yelp
> > > > > Model  |     ppl  |   Rec       | KL   | KLg       | MI |
> > > > > :------:  | :------: |     :------:  | :------: |  :------: | :------: |
> > > > > GPT-2 | 23.40 |- | - | - | -|
> > > > > iVAE | 36.88 | 348.7 | 11.60 | - | 11.00
> > > > > Optimus ($\lambda=0.05$)| 21.99 |  334.31 | 3.09 | - | 2.54
> > > > > Optimus ($\lambda=1.0$)| 24.59 |  325.77 | 27.89 | - | 9.13
> > > > > HSN | **21.70** | 311.18 | 26.67 | 0.31 | **24.93**
> > > > >
> > > > > where KL denotes the Kullback-Leibler divergence between prior and posterior distribution over the local latent variables (i.e. continuous ones for VAE and random walks for HSN). KLg denotes instead the Kullback-Leibler divergence between prior and posterior random graph models in HSN. We also report two results from Optimus, which correspond to their best results as regards MI (between data and latent code) and ppl.
> > > > >
> > > > > HSN outperforms Optimus ($\lambda=1.0$) wrt both MI and ppl in all datasets. HSN also performs comparably to Optimus ($\lambda=0.05$) wrt ppl in Yahoo, while it outperforms all baselines wrt both ppl and MI in the Yelp dataset.
> > > > >
> > > > > These results empirically show HSN is able to infer and interpret schema networks from natural language datasets. Note we used $R=500$ random walk samples and $S=1$ random graph samples, per each sentence in the test set of the datasets. The reported HSN results correspond to schema networks of $K=50$ symbols and random walks of length $L=20$.
> > > > >
> > > > > We hope these results will make the reviewer change back his score to accept.

---

### Author Response · Authors · 2022-12-12
**General comment to all reviewers**

We would like to once more thank all reviewers for their valuable and helpful comments and criticism. We have carefully read and addressed all of them in the discussions below.

We noticed, however, that a common concern among the reviewers is that our results are limited to language modelling tasks. We also noticed that the claim in the abstract might give the impression that our main goal was language modelling. As we expatiate in the **2nd response to Reviewer Dats** below, language modelling represents neither our end goal nor our main result.

Our central goal is instead to impose, via inductive biases, explicit relational structures which allow for compositionality onto the output representations of large, pretrained language models (LPLM). We introduced HSN to achieve this goal.

Indeed, in terms of the HSN modelling assumptions, the task of imposing such explicit structures onto LPLM representations translates into inferring the posterior distributions over both, the global random graph model, and the local random walk model, conditioned on the LPLM representations, that best describe the datasets in question.

Very importantly, note that applying our methodology to commonsense knowledge generation tasks allowed us to introduce a novel notion of reasoning, as random walk processes running along a global semantic graph.  We refer the reader to **2nd response to Reviewer Dats** below, for a detailed exposition of our contributions.

We now briefly highlight the modification we have made as response to the reviewers’ comments and criticisms:

1. We have recomputed the perplexity of HSN via an "importance weighted" Monte Carlo estimator of the marginal log-likelihood. These results can be found in the **On perplexity calculations** reply to reviewer tUks below.

2. Regarding the interpretability of HSN, we have added a new section (Appendix F) in which we explore what symbols the HSN decoder attends to when processing specific words, as to infer the (human-friendly) meaning HSN encodes into the schema network.

3. Regarding concerns about biassed criticism on LPLM, we have reformulated several sentences in the introduction of our work. See **Response to Reviewer nMiy** for details.

4. Regarding clarity concerns, we have added a pseudo-code of our algorithm in Appendix C.

Finally, we plan to

* rewrite the final sentence in the abstract, as to avoid conveying that our main goal is language modelling,
* include, in the Commonsense Reasoning Generation section, additional results on a second dataset: ConceptNet (R. Speer et al., 2017). These results are in the same lines as those preliminary results on the Atomic dataset. See point 3 in **2nd response to Reviewer Dats** below for details.

**Links to references**
1. R. Speer et al., 2017: https://conceptnet.io/

---

### Decision · Program_Chairs · 2023-01-20

**Decision:**

Reject

**Justification For Why Not Higher Score:**

There were too many issues with this paper, and though some of them may have been resolved, the general thrust of the paper was misleading to begin with and I don't think we can accept this paper without another round of review.

**Justification For Why Not Lower Score:**

n/a

**Metareview: Summary, Strengths And Weaknesses:**

This paper describes a new method for sentence representation that uses a latent schema network to back its embeddings.  Through a variety of experiments the authors show that their method can successfully reconstruct such networks from text data alone.

The authors were interested by the new take on representation learning and felt the potential for interpretability was strong.  The enjoyed the idea of a latent schema network and think  approaches using such networks may be promising.

This paper raised some interesting debate amongst reviewers and the authors.  There was a fair amount of disagreement, and two of the most confident reviewers raised very good points.  Significant issues concerned perplexity calculations (which may have been resolved), the intent behind model comparisons and reaching SOTA (which may or may not have been the goal of the paper depending on how previous versions of the abstract and intro were interpreted).  There were also questions about compositionality in related work, which the introductory paragraphs may have misrepresented. During the review period the goals of the paper were rewritten several times, clouding the purpose of the paper and making it more difficult to properly evaluate.  It is clear that the introduction to this paper needs to be reworked so that the goals are laid out more logically, that the experiments are well connected to those goals, and that the experiments do indeed support the claims of the paper.



**Summary Of Ac-Reviewer Meeting:**

This was a borderline paper, but after emailing to set up a meeting, one reviewer changed their score (8->3) making it no longer borderline.  However in response to an author comment they changed their rating yet again (3->8).  At this point I'm not sure we can take this review seriously.  Thankfully this paper had 4 reviews, so discarding this waffling reviewer still leaves 3 reviews.  The most senior reviewers (and most confident reviewers) both give a 3 and raise very valid points.  One other junior reviewer gave an 8, but wasn't willing to champion the paper.